# TMPRSS11E-mediated TFR1 cleavage influences IFN-γR2 internalization and the macrophage innate response
Ting Wang[1], Zhenfa Chen[1], Yiwei Jiang[1], Nannan Wang[2], Wei Zhang[1], Xihua Wang[3], Jie Ding[1], Ling Liu[4], Zichun Hua [5,6,7] ✉, Lei Fang [2] ✉ & Shufeng Li [1] ✉

TMPRSS11E is a serine protease whose expression is upregulated in macrophages during inflammation. Here, we identify TFR1 as an interacting protein of TMPRSS11E via LC-MS/MS. In vitro experiments reveal that TMPRSS11E cleaves TFR1 and releases soluble TFR1 (sTFR1). In alveolar macrophages isolated from pneumonia patients and inflammatory animal models or cultured LPS-challenged cell lines, upregulated TMPRSS11E expression and significantly increased sTFR1 release are observed. Moreover, THP-1 cells stably expressing TMPRSS11E present decreased iron uptake, increased cell surface IFN-γR2 accumulation, and a stronger response to IFN-γ stimulation. During M0 macrophage differentiation to the pro-inflammatory M1 phenotype, the specific induction of TMPRSS11E, decreased cell surface TFR1, and increased IFN-γR2 cell membrane localization are also observed. Taken together, our results suggest that TMPRSS11E contributes to M1 macrophage differentiation by regulating iron uptake and affecting IFN-γR2 internalization through TFR1 cleavage, indicating that TMPRSS11E plays an important role in iron homeostasis and the innate immune response.

TMPRSS11E, also called DESC1, is a type II transmembrane serine protease. These proteases contain a catalytic domain at the carboxyl-terminal region. This domain possesses a catalytic triad of three amino acids (serine, aspartate, and histidine) that is essential for catalytic activity[1]. Previous data suggested that TMPRSS11E is specifically expressed in epithelial cells and plays a role in the development and progression of some types of tumors[2,3]. Our recent research revealed that TMPRSS11E expression is significantly induced by LPS stimulation in alveolar macrophages in vivo and that upregulated TMPRSS11E expression further aggravates the inflammatory response[4]. The public GEO datasets GDS5356 and GDS3549 also revealed that in the RAW264.7 macrophage line, TMPRSS11E expression is induced in response to murine norovirus infection. Owing to the important role of TMPRSS11E in macrophage responses to infection, TMPRSS11E-interacting proteins were investigated. Coimmunoprecipitation coupled with mass spectrometry was subsequently performed. Transferrin receptor protein 1 (TFR1) was identified as a binding protein of TMPRSS11E. We

subsequently performed biochemical experiments to confirm the interaction between TMPRSS11E and TFR1. Further validation revealed that TFR1 is a TMPRSS11E substrate and can be cleaved by TMPRSS11E. TFR1 is a membrane glycoprotein that is responsible for cellular iron uptake by binding plasma transferrin (Tf) and is involved in iron homeostasis[5,6].

To date, TFR1 cleavage in macrophages has not been investigated[7,8], and the consequences and significance of downregulated TFR1 expression in macrophage function and the immune response are also unknown. In this study, in alveolar macrophages from the BALF of patients with clinical pneumonia and an LPS-challenged mouse model, TMPRSS11E was induced, and a significantly increased concentration of sTFR1 was detected. Moreover, under LPS stimulation, through the cleavage of TFR1, TMPRSS11E modulated macrophage iron uptake and influenced membrane IFN-γR2 internalization; therefore, TMPRSS11E plays an important role in the IFN-γ-STAT1 pathway response and proinflammatory polarization.

[1]Key Laboratory of Developmental Genes and Human Disease in Ministry of Education, Jiangsu Provincial Key Laboratory of Critical Care Medicine, Department of Biochemistry and Molecular Biology, Medical School of Southeast University, Nanjing, China. [2]Jiangsu Key Laboratory of Molecular Medicine, Chemistry and Biomedicine Innovation Center, Medical School of Nanjing University, Nanjing, China. [3]Department of Respiratory of Zhongda Hospital, Southeast University, Nanjing, China. [4]Jiangsu Provincial Key Laboratory of Critical Care Medicine, Department of Critical Care Medicine, Zhongda Hospital, Medicine School of Southeast University, Nanjing, China. [5]State Key Laboratory of Pharmaceutical Biotechnology, School of Life Sciences, Nanjing University, Nanjing, Jiangsu, China. [6]Changzhou High-Tech Research Institute of Nanjing University and Jiangsu TargetPharma Laboratories Inc, Changzhou, China. [7]Faculty of Pharmaceutical Sciences, Xinxiang Medical University, Xinxiang, China. ✉e-mail: huazc@nju.edu.cn; njfanglei@nju.edu.cn; shufengli@seu.edu.cn

## Results

### TMPRSS11E interacts with TFR1

To identify the potential substrate of TMPRSS11E, pCMV-TMPRSS11E-Flag was transfected into HEK293T cells (Supplementary Table 1). TMPRSS11E complexes from these cells were purified with anti-Flag-conjugated beads and then resolved by SDS–PAGE (Supplementary Fig. 1). The resolved protein complexes were digested, and the digested peptides were then analyzed by MS/MS (Fig. 1A–C). The results illustrated that TFR1 interacted with TMPRSS11E. To validate the interaction between TMPRSS11E and TFR1, immunoprecipitation (IP) and immunoblotting were performed. An association between TMPRSS11E and TFR1 was observed after HEK293T cells were co-transfected with pCMV-TMPRSS11E-Flag and pCMV-V5-TFR1-myc (Fig. 1D). Together, our results revealed that TMPRSS11E and TFR1 interact, establishing the basis for further exploration of TFR1 cleavage by TMPRSS11E.

### TFR1 cleavage by TMPRSS11E

Full-length TFR1 is expressed on the cell surface and plays a crucial role in mediating iron homeostasis. Extracellular ferric ions ($Fe^{3+}$) bind transferrin (Tf) and are transported via the TFR1 on the cell membrane. First, the transferrin/TFR1 complexes are endocytosed into the cell. Subsequently, in the endosome, $Fe^{3+}$ dissociates from Tf and is reduced to $Fe^{2+}$ by STEAP and

transported by DMT1 to the cytoplasm. In addition to full-length membrane TFR1, the truncated form of soluble TFR1 (sTFR1) is produced by proteolytic cleavage between arginine (100) and leucine (101)[9], after which sTFR1 is shed and released[10]. To date, the significance of TFR1 cleavage is still not known. Whether macrophage TFR1 is cleaved to release sTFR1 has not been investigated until now.

To assess whether TMPRSS11E is involved in TFR1 cleavage, first, purified recombinant TMPRSS11E and recombinant Fc-TFR1 proteins were incubated for 30 minutes (Supplementary Table 2), then western blot analysis was performed (Fig. 2A). Representative immunoblot images and results of densitometry analysis of blots (Supplementary Fig. 2A) revealed that TMPRSS11E cleaved TFR1. Moreover, the ability of TMPRSS11E to cleave cell surface TFR1 was also tested. Western blot analysis revealed that the recombinant TMPRSS11E protein cleaved cell surface TFR1, releasing a large amount of the 75 kDa soluble TFR1 (sTFR1) into the cell medium (Fig. 2B, Supplementary Fig. 2B).

To investigate whether the proteolytic activity of TMPRSS11E is necessary for TFR1 cleavage, pCMV-GFP-TMPRSS11E and pCMV-GFP-TMPRSS11E S372A were constructed (Fig. 2C). Like other TTSPs, TMPRSS11E was synthesized as an inactive single-chain zymogen. The activation of TMPRSS11E requires proteolytic cleavage at a conserved activation site that converts the enzyme from a one-chain zymogen to an

**A**

| Acc. | IgG | | IP | | Fold Change | Protein Name |
|------|-----|-----|-----|-----|-------------|--------------|
| | Num Unique | Peptide Count | Num Unique | Peptide Count | | |
| Q9UL52 | 0 | 0 | 24 | 217 | NA | Transmembrane protease serine 11E |
| P02786 | 2 | 2 | 8 | 9 | 4.5 | Transferrin receptor protein 1 |

**B**

| TFR1(14.6%Cov) | | |
|----------------|------|---|
| Peptide sequence | m/z | z |
| GFVEPDHYVVVGAQR | 558.2944 | 3 |
| LAQMFSDMVLK | 641.8277 | 2 |
| LTHDVELNLDYER | 539.6045 | 3 |
| LVYLVENPGGYVAYSK | 886.4646 | 2 |
| SAFSNLFGGEPLSYTR | 873.429 | 2 |
| SSGLPNIPVQTISR | 734.9079 | 2 |
| VEYHFLSPYVSPK | 522.6069 | 3 |
| VSASPLLYTLIEK | 717.4156 | 2 |

**C**

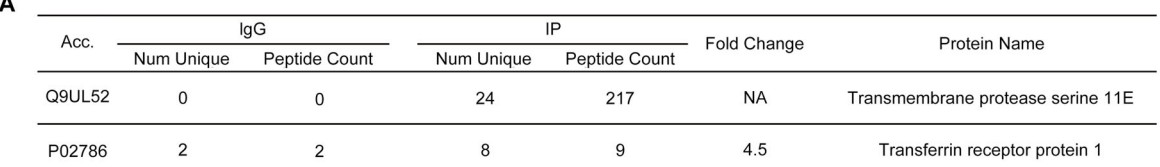

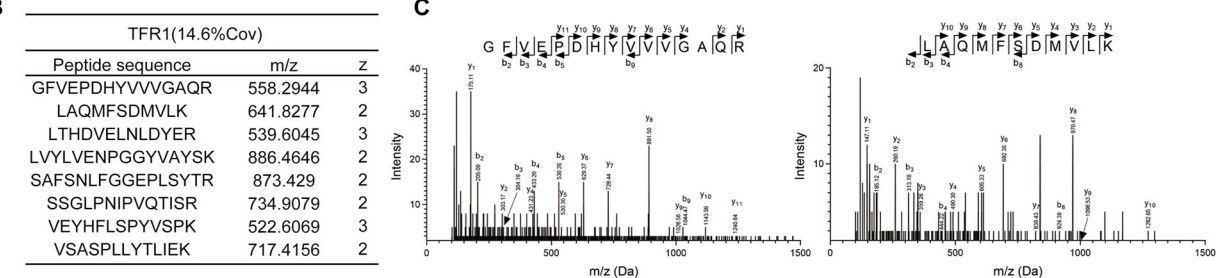

**D**

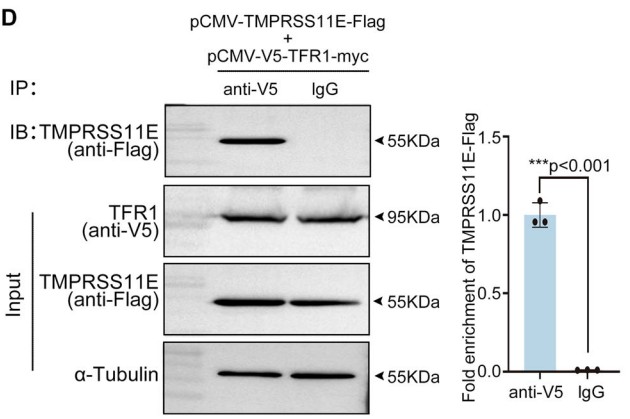

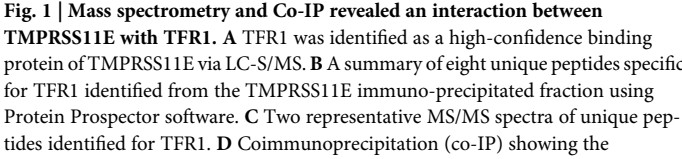

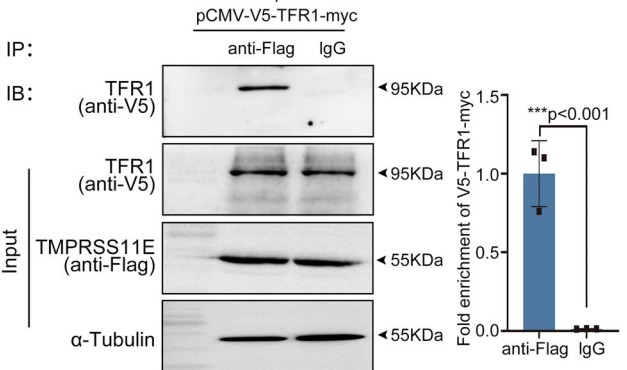

**Fig. 1 | Mass spectrometry and Co-IP revealed an interaction between TMPRSS11E with TFR1. A** TFR1 was identified as a high-confidence binding protein of TMPRSS11E via LC-S/MS. **B** A summary of eight unique peptides specific for TFR1 identified from the TMPRSS11E immuno-precipitated fraction using Protein Prospector software. **C** Two representative MS/MS spectra of unique peptides identified for TFR1. **D** Coimmunoprecipitation (co-IP) showing the

interaction between TMPRSS11E and TFR1. pCMV-TMPRSS11E-Flag and pCMV-V5-TFR1-Myc were cotransfected into HEK293T cells, and the expressed fusion proteins were then precipitated with anti-V5 or anti-Flag antibodies, respectively. After the co-IP of TMPRSS11E-Flag and V5-TFR1-myc, western blotting was performed. Statistical analysis of immunoprecipitation enrichment was shown (*n* = 3).

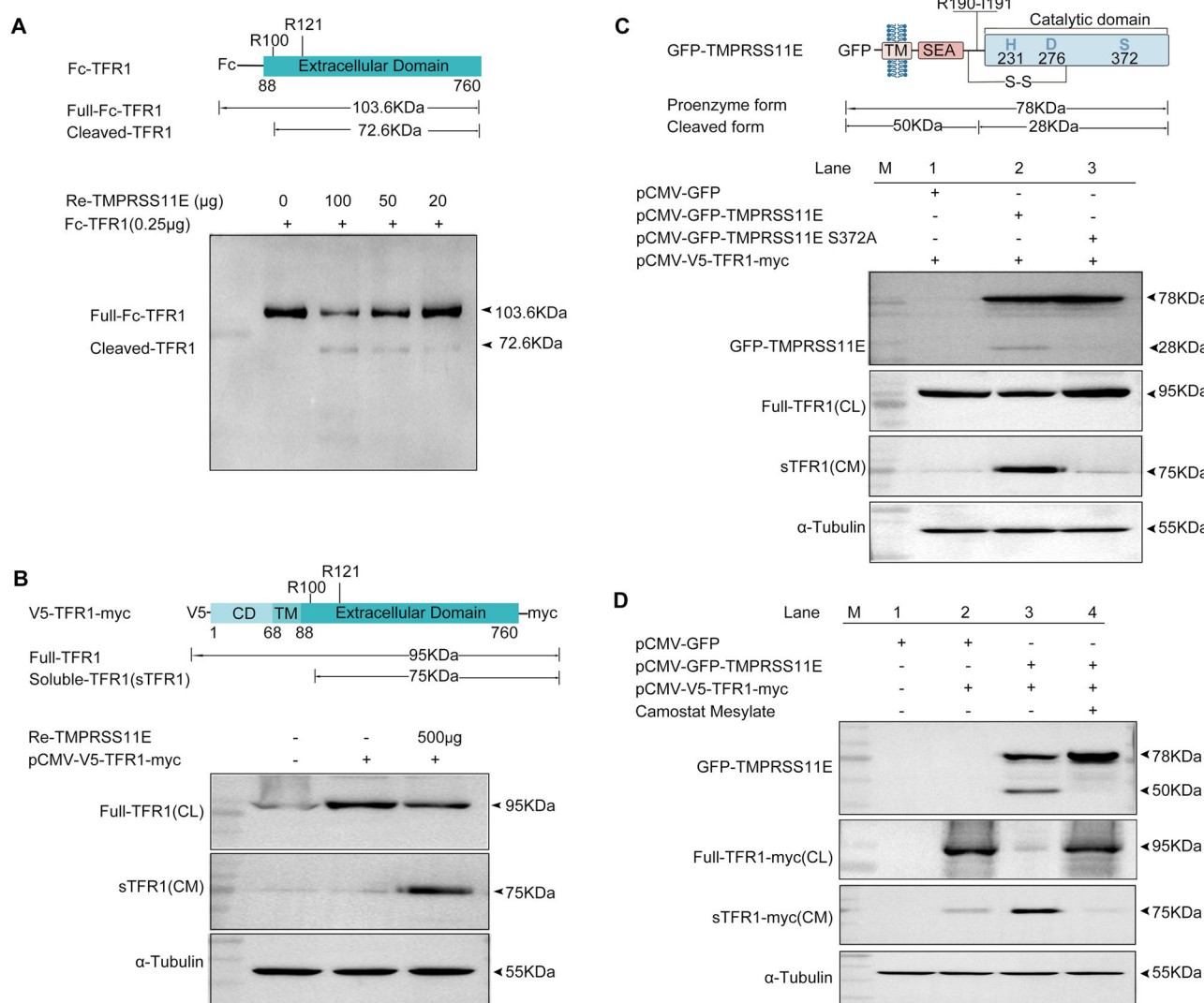

**Fig. 2 | TMPRSS11E cleaves TFR1 and releases soluble TFR1. A** Western blot analysis of the cleavage of recombinant Fc-TFR1 by TMPRSS11E recombinant protein. A schematic diagram of recombinant Fc-TFR1 is shown. The Fc-tag, the extracellular domain of TFR1, the supposed TMPRSS11E-processing sites, and the predicted molecular weight of Fc-TFR1 are also indicated. **B** The cleavage of cell surface TFR1 by recombinant TMPRSS11E was analyzed by western blot. Schematic representation of the TFR1 expression construct pCMV-V5-TFR1-myc. Depicted are the cytoplasmic domain (CD), transmembrane domain (TM), extracellular domain, N-terminal V5-tag, and C-terminal myc-tag. Twenty-four hours after transfection of HEK293T cells, TMPRSS11E protein was added for incubation. Then, the conditioned medium (CM) and cell lysate (CL) were prepared for western blot analysis. **C** Effect of TMPRSS11E mutation on the cleavage of TFR1. The GFP-

tagged full-length TMPRSS11E construct pCMV-GFP-TMPRSS11E is shown. TM transmembrane, SEA domain, the cleavage site for proteolytic activation, the catalytic residues in the serine protease domain, and disulfide bonds (s-s) are indicated. The expected molecular masses of the TMPRSS11E zymogen and the activated, cleaved protease catalytic domain are also shown. HEK293T cells were co-transfected with the indicated plasmids, cell lysates and 24-h conditioned medium were analyzed by western blotting. **D** Effect of TMPRSS11E inhibitor on the cleavage of TFR1. After transfection of 24 h, cells were incubated with TMPRSS11E inhibitor or control; then the conditioned medium and cell lysates were prepared for western blot analysis. Statistical analysis of protein level in western blot images was shown in Fig.S2 ($n = 3$).

active, two-chain protease[11,12] (Fig. 2C). Our previous research revealed that TMPRSS11E activation can be mediated by autocatalysis[3,4]. The results showed that in pCMV-GFP-TMPRSS11E-transfected cells, a prominent band with a molecular mass roughly expected for the GFP-tagged zymogen form (78 kDa) and a catalytic fragment (28 kDa) was observed. Moreover, following cotransfection of the pCMV-GFP-TMPRSS11E- and TFR1-expressing plasmids, sTFR1 was detected in the medium. In contrast, in pCMV-GFP-TMPRSS11E S372A and TFR1-expressing plasmids cotransfected group, S372A mutant abolished the catalytic function of TMPRSS11E[3,4]; no 28 kDa band corresponding to the catalytic TMPRSS11E fragment was detected, and TFR1 cleavage was also abrogated (Fig. 2C lane3), meanwhile the cell surface TFR1 level (Full-TFR1-myc(CL)) was

significantly greater than that in lane 2 of pCMV-GFP-TMPRSS11E transfected group, reflecting the impaired cleavage of mutated TMPRSS11E (Supplementary Fig. 2C).

To characterize sTFR1 release from membranes, we also assessed the influence of TMPRSS11E protease inhibitor on sTFR1 release. Figure 2D shows that in wild-type TMPRSS11E-transfected cells, TMPRSS11E auto-activation resulted in 50 kDa GFP-tagged N-terminal fragments under reducing conditions (lane 3), as determined using anti-GFP antibodies; the release of sTFR1 was significantly elevated in the culture medium, this finding was consistent with decreased surface TFR1(lane 3). In contrast, when treatment with the TMPRSS11E inhibitor (camostat mesylate) for 20 h, the autocleavage and catalytic functions of TMPRSS11E were

**Fig. 3 | TMPRSS11E cleaves TFR1 at R100.**
**A** Schematic representation of TFR1 mutant-expressing constructs. **B**, **C** Western blot analysis of lysates and conditioned medium from HEK293T cells or A549 cells cotransfected with pCMV-GFP-TMPRSS11E and pCMV-V5-TFR1-myc or mutant constructs. The expression of the transfected pCMV-GFP-TMPRSS11E protein was detected under non-reducing conditions with an anti-TMPRSS11E antibody. Loading was estimated with an anti-tubulin antibody. Quantification of protein level in Western blot images is shown. Data are expressed as mean ± SD ($n = 3$).

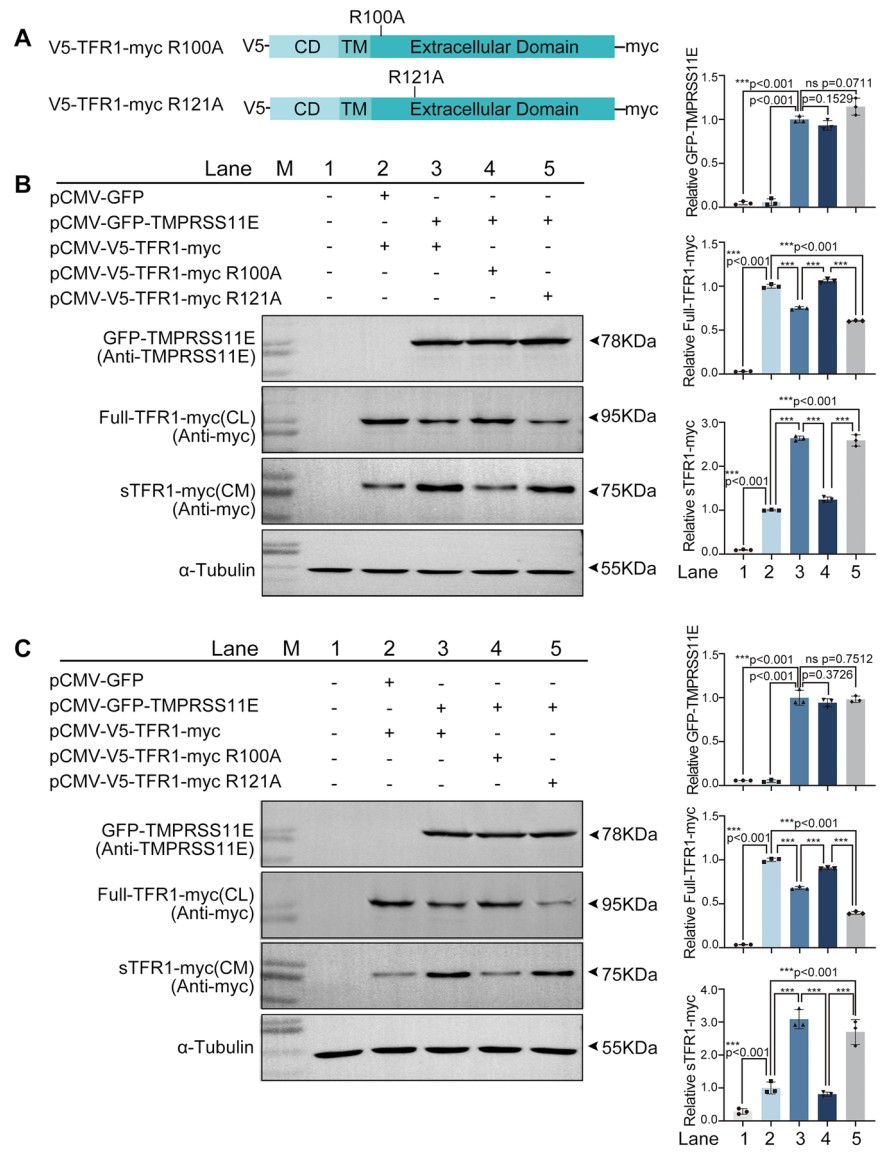

inhibited, as evidenced by no 50 kDa band of GFP-tagged N-terminal fragments (lane 4). The shedding of sTFR1 in the culture medium was also inhibited (Fig. 2D lane 4, Supplementary Fig. 2D). Taken together, these results indicate that TFR1 is a TMPRSS11E substrate.

### TFR1 cleavage at R100 by TMPRSS11E

Because TMPRSS11E cleaves substrates at basic residues, in the present study, to better characterize the release of sTFR1 by TMPRSS11E, a basic amino acid in the proposed cleavage site (R100) of TFR1, as well as the downstream putative site Arg121, were mutated, and TFR1 mutants were constructed (Fig. 3A). Then, HEK293T cells and A549 cells were cotransfected with TFR1 or mutant plasmids together with pCMV-GFP-TMPRSS11E, and the resulting cell lysates and conditioned media were analyzed by immunoblotting (Fig. 3B, C). The expression of TMPRSS11E or TFR1 was confirmed in cell lysates, and more sTFR1 was observed in the culture medium of the pCMV-V5-TFR1-myc and pCMV-V5-TFR1-myc R121A groups. These results indicated that the R121A mutation did not influence TFR1 cleavage. When pCMV-V5-TFR1-myc R100A was cotransfected with TMPRSS11E, obviously less sTFR1 was detected, which suggested that the R100A mutation impaired surface TFR1 cleavage greatly. All the sTFR1 detected in the culture medium was 75 kDa. These results suggest that TMPRSS11E cleaves TFR1 at R100.

### More sTFR1 is released when TMPRSS11E expression is upregulated in the alveolar macrophages in BALF from pneumonia patients and LPS-treated rats

To better understand the pathological significance of TMPRSS11E and TFR1 in human macrophages under inflammatory conditions, we first collected alveolar macrophages from the BALF of pneumonia patients and control patients and assessed the expression of TMPRSS11E (Supplementary Table 3). The immunohistochemistry results revealed that TMPRSS11E was increased in alveolar macrophages from pneumonia patients (Fig. 4A1). The concentrations of sTFR1 in BALF were significantly increased (Fig. 4A2). Our results are consistent with a previous report that the concentration of sTFR1 in bronchoalveolar lavage fluid is significantly increased in acute respiratory distress syndrome patients[13]. These findings suggest that increased TMPRSS11E disrupts normal surface TFR1 cleavage and iron metabolism in the lungs of acute respiratory distress syndrome patients and pneumonia patients.

Next, we evaluated the relationship between TMPRSS11E and sTFR1 in an LPS-treated animal model. In practice, both the LPS lung inflammation rat model and mice model were widely used[14,15]. In vitro models of lung inflammation usually consist of stimulation of isolated pulmonary cells with respective pathogens or LPS. In practice, isolation of alveolar macrophage from mice is a relatively simple procedure that only requires

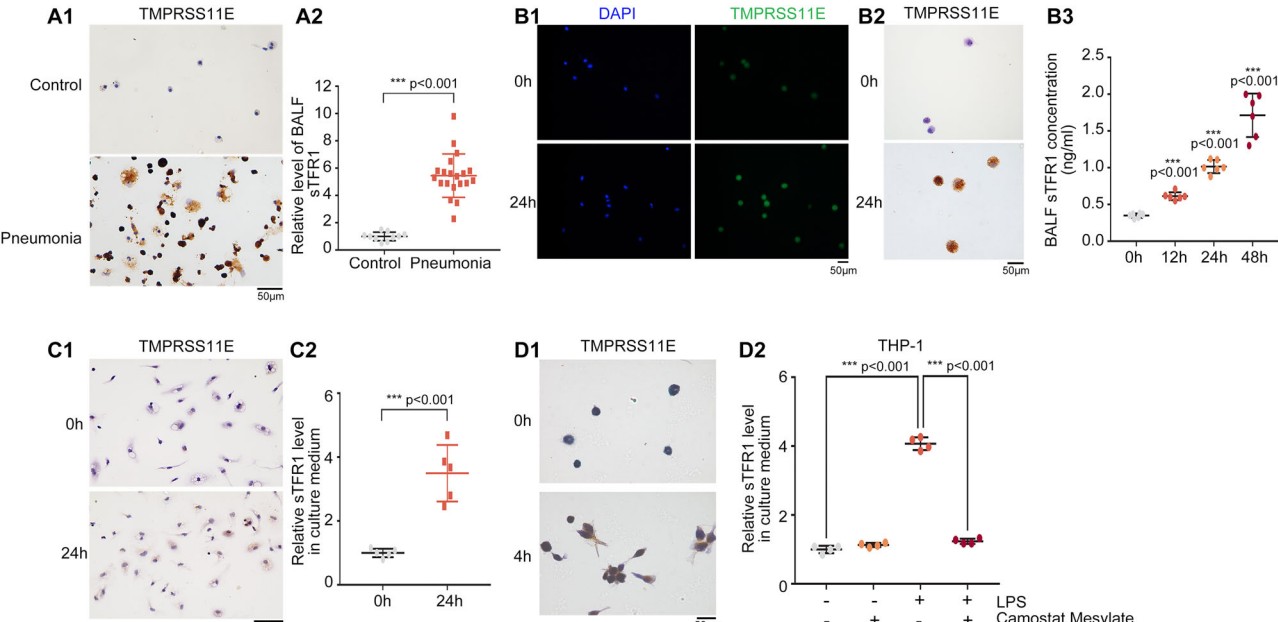

**Fig. 4 | During inflammation, TMPRSS11E expression is induced in macrophages, and the sTFR1 level also increases. A** Alveolar macrophages were obtained from the BALF of pneumonia patients ($n = 20$) or control patients ($n = 10$). A1. Immunocytochemistry staining of TMPRSS11E. Scale bars, 50 μm. A2. The sTFR1 level in BALF was also measured. **B** Alveolar macrophages from the BALF of control or LPS-treated rats. B1. Immunofluorescence staining of TMPRSS11E. B2. Immunocytochemistry staining of TMPRSS11E. Scale bars, 50 μm. B3. Quantification of

sTFR1 in BALF by ELISA ($n = 6$). **C** Mouse peritoneal macrophages were collected and treated with LPS for the indicated time. C1. Immunocytochemistry staining of TMPRSS11E. C2. Quantification (ELISA) of sTFR1 in the medium of cultured LPS-treated mouse peritoneal macrophages ($n = 5$). **D** THP-1 cells were cultured and treated with LPS. D1. Immunocytochemistry staining of TMPRSS11E. Scale bars, 50 μm. D2. Quantification (ELISA) of sTFR1 in medium from cultured LPS-treated THP-1 cells ($n = 4$).

performing a bronchoalveolar lavage in mice. For low yield of mice bronchoalveolar lavage, which almost invariably requires many animals to obtain enough numbers of cells. While BALF from rats could provide many more alveolar macrophages[16,17]. Therefore, BALF was collected from LPS-treated rats. Immunofluorescence and IHC staining revealed that the expression of TMPRSS11E was increased in alveolar macrophages from the BALF of LPS-treated rats (Fig. 4B1, B2). Moreover, soluble TFR1 was also detected in the supernatant of BALF, and increased sTFR1 levels were detected in the supernatant of BALF from LPS-treated rats than from control rats (Fig. 4B3). These results suggest that under inflammatory conditions, surface TFR1 cleavage in rats' increases.

Moreover, under inflammatory conditions, increased TMPRSS11E in mouse alveolar macrophages was also confirmed. LPS-treated mouse models were generated, and lung tissues were obtained. Western blotting of whole-lung homogenates revealed that the protein levels of TMPRSS11E in the LPS-treated group were significantly greater than those in the control group (Supplementary Fig. 3A). Pathological changes in lung tissues are shown in Supplementary Fig. 3B. H&E staining of the lung tissues showed thickened alveolar walls, less alveolar cavity, and inflammatory cell infiltration in the LPS group. It was also found that MPO level was significantly elevated in the LPS-treated group which reflect neutrophil accumulation in the lungs (Supplementary Fig. 3C). Notably, immunofluorescence staining revealed increased TMPRSS11E expression in the lung tissue of mice in the LPS-treated group (Supplementary Fig. 3D). Immunohistochemical staining of the macrophage marker CD68 as well as TMPRSS11E in the lung tissues of the mice confirmed strong positive TMPRSS11E immunoreactivity in the lung macrophages of LPS-treated mice (Supplementary Fig. 3E). These data indicate that LPS-induced lung inflammation is characterized by an increase in TMPRSS11E primarily in macrophages. Together, these data suggest that increased TMPRSS11E in macrophages contributes to increased TFR1 cleavage and increased sTFR1 in BALF during inflammation.

### TMPRSS11E expression is upregulated in LPS-treated peritoneal macrophages and THP-1 cells

In addition, primary peritoneal macrophages are also widely used for studies of macrophage inflammation[18–21]. To characterize the effects of TMPRSS11E on increased sTFR1 levels in macrophages cultured in vitro, mouse peritoneal macrophages were isolated and treated with LPS for 24 h. IHC staining revealed that after LPS treatment, TMPRSS11E expression was upregulated, and sTFR1 was also increased (Fig. 4C). In addition, THP-1 monocytes were cultured and differentiated into macrophages using PMA, and then, THP-1-derived macrophages were cultured in the presence of LPS and analyzed by IHC. The results revealed that TMPRSS11E expression was upregulated after LPS treatment (Fig. 4D1). In addition, sTFR1 was also increased after LPS treatment, and the increase in sTFR1 was reversed when TMPRSS11E inhibitor was present, suggesting that increased TMPRSS11E resulted in increased sTFR1 release during inflammation. Taken together, our results indicate that TMPRSS11E cleaves TFR1 in inflammatory macrophages and may further influence macrophage immune function.

### Cell surface TFR1 and IFN-γR2 in stable THP-1-TMPRSS11E cell lines

To investigate the function and significance of TMPRSS11E in macrophages, the stable cell lines THP-1-TMPRSS11E and THP-1-TMPRSS11E S372A were generated to stably express TMPRSS11E or the TMPRSS11E S372A mutant. The influence of TMPRSS11E expression on cell surface TFR1 was examined. Tf incorporation is positively correlated with the cell membrane TFR1 level[22]. Stable cell lines were incubated with Alexa-488-conjugated Tf, and the mean fluorescence intensity (MFI) of stable THP-1-TMPRSS11E cells was much lower than that of control cells or stable THP-1-TMPRSS11E S372A cells, indicating decreased Alexa-488-labeled Tf uptake and lower surface TFR1 (Fig. 5A1, A2). Western blot analysis confirmed that in stable THP-1-TMPRSS11E cells, the TFR1 level was decreased. In stable THP-1-TMPRSS11E S372A cells, TMPRSS11E activity was absent, and the TFR1 level was restored (Fig. 5B). Moreover, when stable THP-1-TMPRSS11E cells were incubated with camostat, the

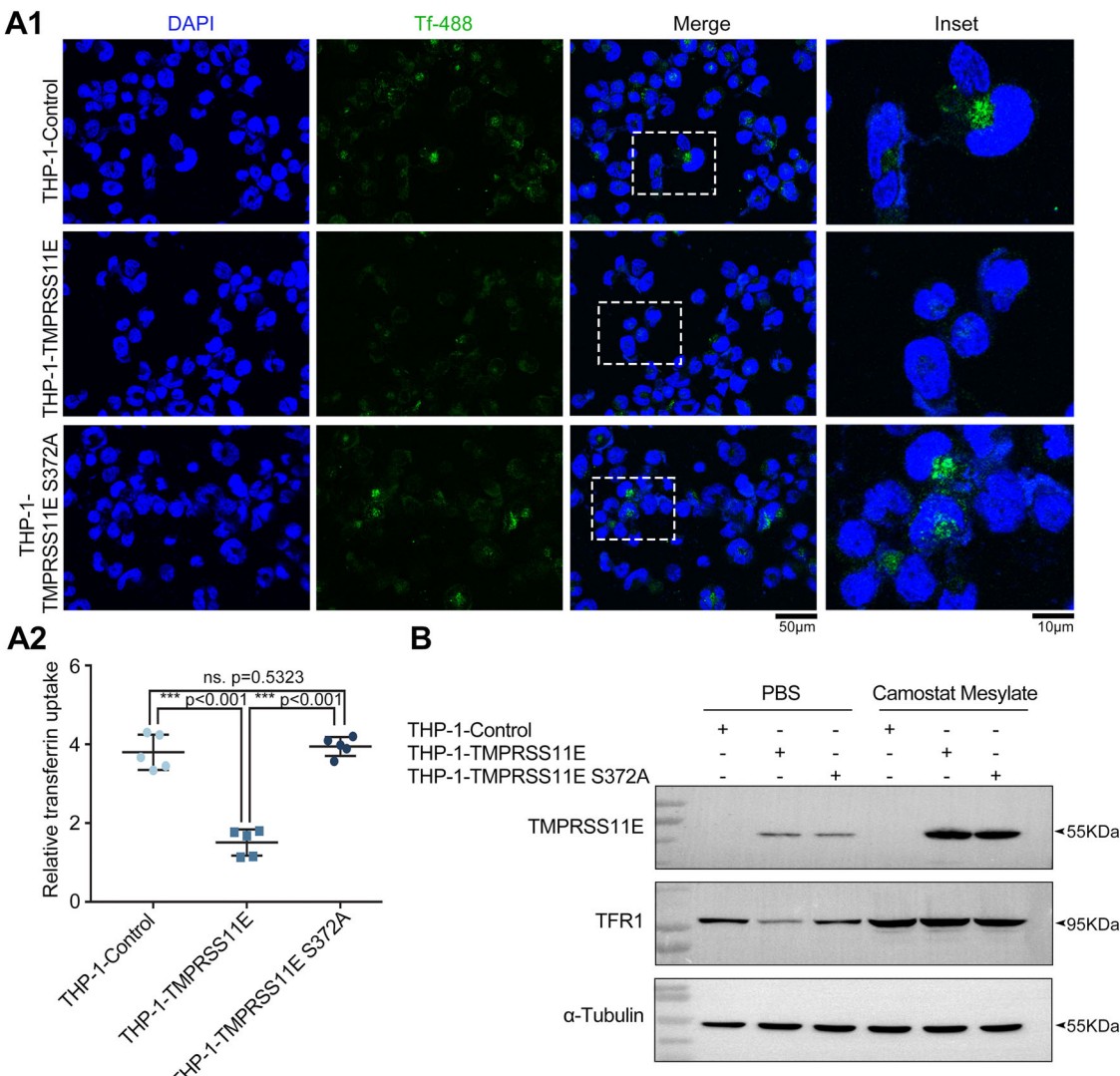

**Fig. 5 | Effect of TMPRSS11E on macrophage surface TFR1 levels. A** THP-1 cells stably overexpressing TMPRSS11E or TMPRSS11E S372A were generated, and the effects of TMPRSS11E or mutant TMPRSS11E S372A overexpression on the cellular TFR1 level and transferrin uptake were analyzed. Stable THP-1-TMPRSS11E cells or stable THP-1-TMPRSS11E S372A cells were incubated with Alexa 488-labeled transferrin (Tf) for 30 minutes at 4 °C. The cells were then transferred to 37 °C for 20 minutes before being returned to ice. After washing with PBS, the cells were fixed and stained with DAPI. Scale bars: 50 μm for the low-magnification images and 10 μm for the high-magnification insets on the right (A1). A2. The results of three independent experiments are presented in (A1). In experimental design, duplicate wells for each condition were performed twice, and in the third time, one well for one condition was repeated ($n = 5$). **B** Effect of TMPRSS11E activity on the TFR1 level. Stable cells were pretreated with camostat mesylate (serine protease inhibitor) or PBS for 30 minutes. The cell lysates were collected and analyzed by western blotting.

proteolytic activity of TMPRSS11E was inhibited and as a result of inhibition, TMPRSS11E could not auto-activated or cut TFR1, the full length TFR1 levels increased (Fig. 5B, Supplementary Fig. 4A). These data suggest that TMPRSS11E expression in macrophages influence surface TFR1 levels. More interestingly, the level of TMPRSS11E in the group receiving camostat treatment also seemed to have increased, maybe reflecting that another endogenous serine protease was inhibited, which might affect TMPRSS11E level. However, this speculation still needs more investigation.

Macrophages are considered important immune effector cells. IFN-γ is considered the principal effector cytokine in immunity and inflammation and exerts its effects on target cells through receptor IFN-γR complex[23]. The IFN-γR complex is composed of two IFN-γR1 and two IFN-γR2 chains. IFN-γR1 is ubiquitous, whereas IFN-γR2 serves as a key regulator of IFN-γ responsiveness and determines the outcome of IFN-γ–STAT1-dependent downstream pro-inflammatory responses[24]. Notably, abnormalities in IFN-γR1 and IFN-γR2 expression are closely associated with Mendelian susceptibility to mycobacterial disease (MSMD), a syndrome characterized by localized or disseminated infections caused by atypical mycobacteria[25].

Interestingly, in T cells, the binding of transferrin with TFR1 induces the internalization of IFN-γR2 and limits IFN-γ–STAT1 signaling[26], and the blockade of TFR1 internalization also decreases IFN-γR2 internalization[26]. Given the pivotal role that TMPRSS11E plays in regulating the level of TFR1 on the macrophage surface, we investigated whether the IFN-γR2 level was also affected by TFR1 and TMPRSS11E in macrophages. For the critical roles of IFN-γR2 in immunity and disease, understanding whether TMPRSS11E and TFR1 mediated iron uptake affects IFN-γR2 internalization in macrophages has important biological significance. IFN-γR2 distribution and internalization markers Rab5a were evaluated by confocal microscopy using stable cell lines (Fig. 6A). Rab5a is a member of the Rab family of small GTPases and plays a crucial role in endocytosis. It is used as a marker of early sorting endosomes[27]. Rab5 was associated with the internalization of receptors, and endocytosis of receptors has been considered as a mechanism of signal attenuation[28]. Our experiments revealed that some IFN-γR2 was present at the cell surface membrane; meanwhile, a substantial fluorescent signal was also located in intracellular endosome-like vesicles (permeabilized cells), suggesting that IFN-γR2 was internalized from the cell

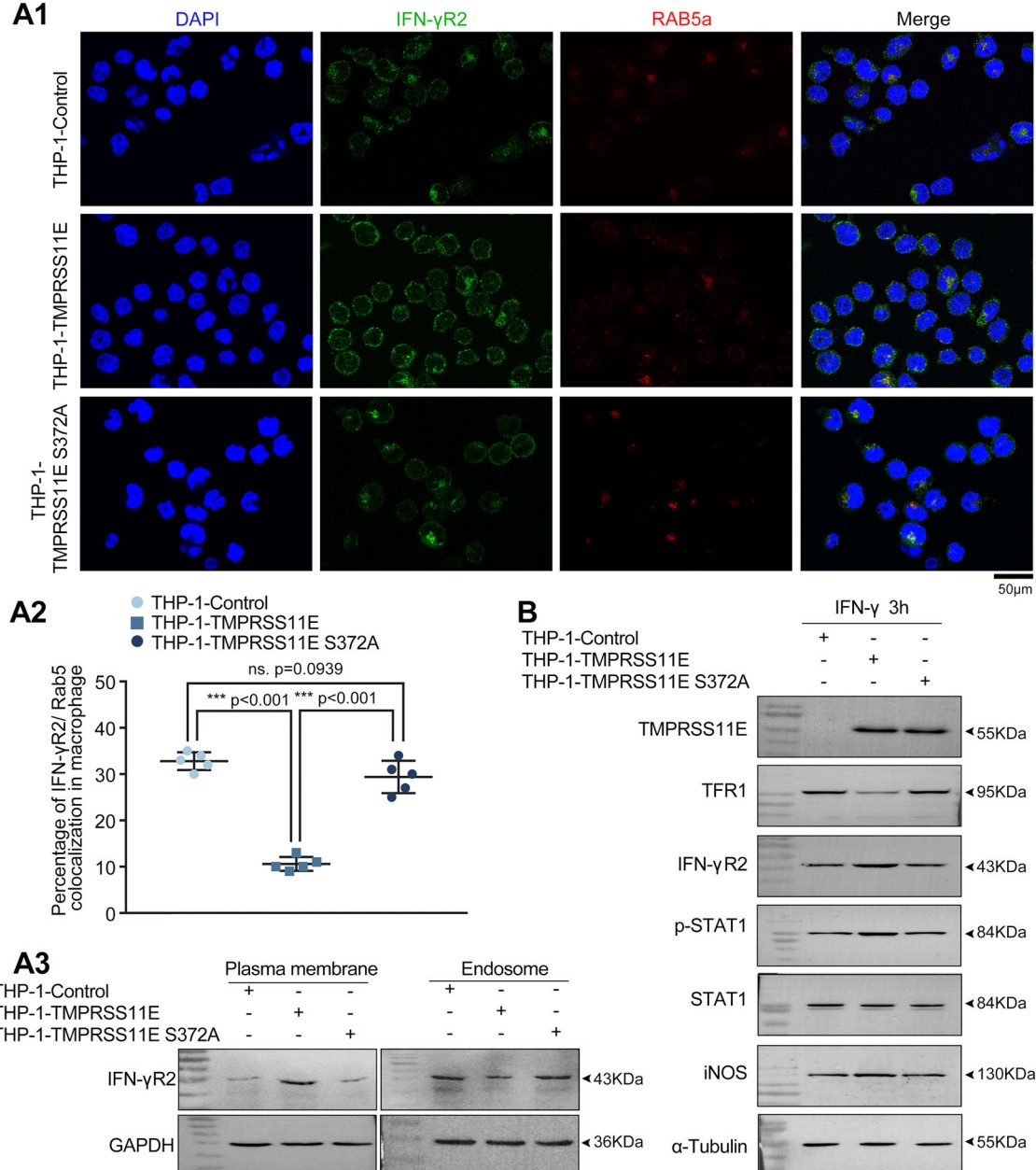

**Fig. 6 | Effect of TMPRSS11E on IFN-γR2 internalization and the IFN-γ signaling response. A** Double fluorescence staining of IFN-γR2 and Rab5a in THP-1-TMPRSS11E cells or control cells. A1. Representative fluorescence images. Scale bars, 50 μm. Rab5a was used to mark endosomes. A2. IFN-γR2 internalization was determined by the colocalization of IFN-γR2 and Rab5a ($n = 5$). A3. Western blot analysis of IFN-γR2 in the isolated endosome or plasma membrane from cultured THP-1-TMPRSS11E stable cells or control cells. GAPDH in the cytosol was used as an internal control. **B** TMPRSS11E-overexpressing cells respond to IFN-γ stimulation more strongly through IFN-γR2-STAT1 pathway activation. THP-1-TMPRSS11E or control cells were stimulated with IFN-γ for 3 h, and the protein levels of p-STAT1, IFN-γR2, and iNOS were analyzed by western blot.

surface into the cytoplasm with a punctate distribution. Enhanced staining of the cell perimeter in THP-1-TMPRSS11E cells than in control cells suggests increased IFN-γR2 on the plasma membrane. The decreased colocalization between IFN-γR2 and Rab5a was observed in THP-1-TMPRSS11E, which indicated that increased IFN-γR2 on the membrane was caused by decreased endocytosis. We also confirmed that in THP-1-TMPRSS11E cells, the IFN-γR2 protein was increased in the isolated cell membrane but decreased in endosome (Fig. 6A3, Supplementary Fig. 4B). Western blot analysis revealed lower TFR1 and higher IFN-γR2 levels and a stronger response to IFN-γ signaling in stable THP-1-TMPRSS11E cells (Fig. 6B, Supplementary Fig. 4C). Our results indicated that TMPRSS11E and cell surface TFR1 levels also influence IFN-γR2 levels in macrophages, which is consistent with the mechanism in T cells[26,29].

## Expression of TMPRSS11E in THP-1-derived polarized macrophages

Macrophages have a high degree of heterogeneity and plasticity and can differentiate into two main types of macrophages. In proinflammatory M1 macrophages, the level of cellular TFR1 is lower than that in anti-inflammatory M2 macrophages[30]; thus, we investigated the role of TMPRSS11E in different macrophage subtypes. First, we used a classic polarization scheme to induce the polarization of THP-1-derived macrophages toward the M1 and M2 phenotypes. In agreement with previous studies, flow cytometry analysis revealed that LPS- and IFN-γ-treated THP-1 macrophages significantly expressed the M1 macrophage marker CD86, whereas IL-4 induced M2 macrophage differentiation, with macrophages highly expressing the M2 marker CD163. The expression of M1 and M2

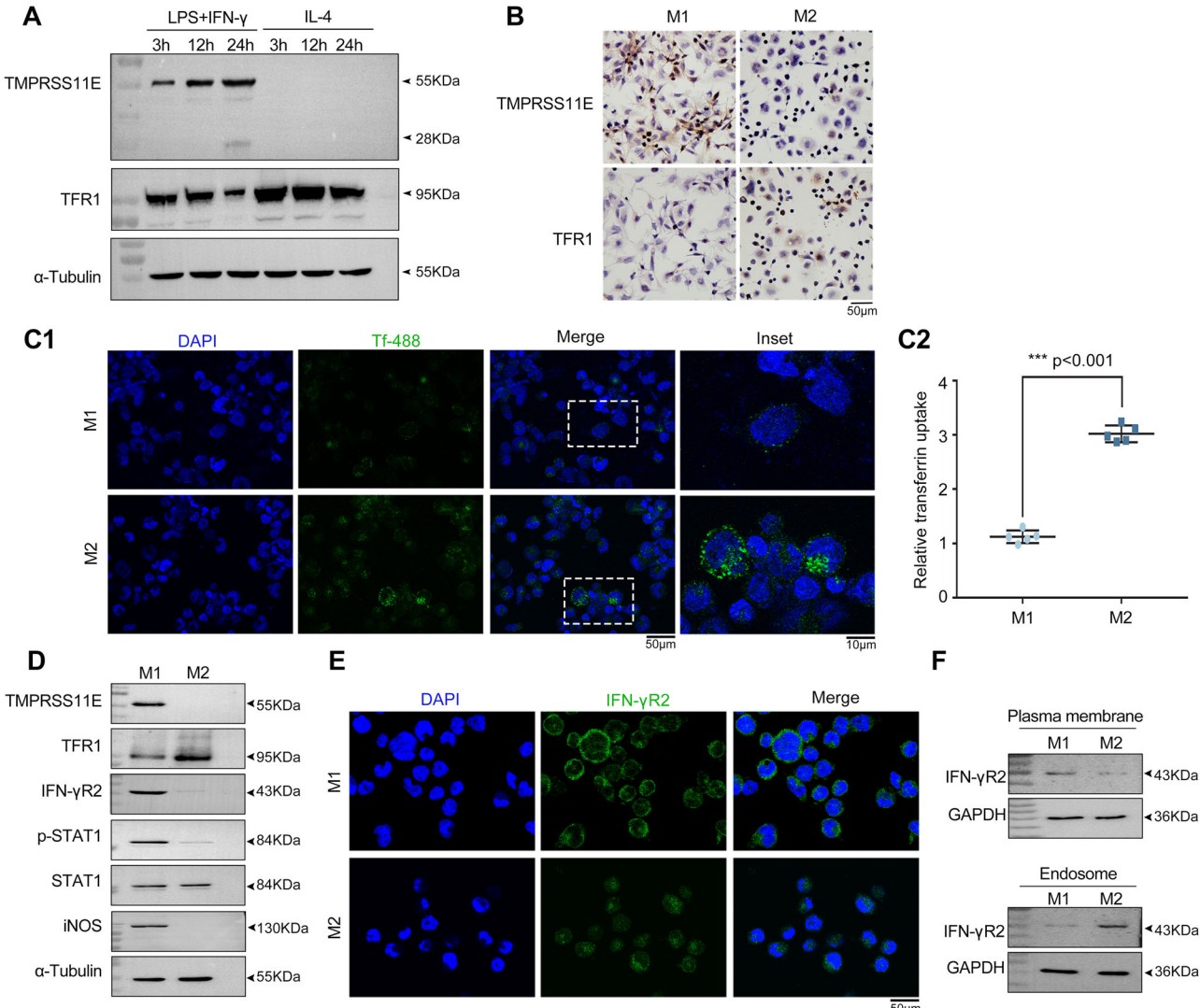

**Fig. 7 | TMPRSS11E expression during M1 polarization. A**. THP-1 cells were stimulated with LPS + IFN-γ or IL-4 for the indicated times, and western blot analysis was performed. **B**. TMPRSS11E or TFR1 expression levels in THP-1-differentiated M1 or M2 macrophages were assessed by immunocytochemistry (*n* = 3). **C** Immunofluorescence analysis of transferrin (Tf) uptake after the addition of Alexa 488-labeled Tf. THP-1 cells were stimulated with LPS + IFN-γ or IL-4 for 24 h, then incubated with the fluorescent dye Alexa 488-labeled transferrin (Tf) for 30 min at 4 °C. The cells were then transferred to 37 °C for 20 min before being returned to ice. After washing with PBS, the cells were fixed, and the nuclei were stained with DAPI. Scale bars: 50 μm for the low-magnification images and 10 μm for the high-magnification insets on the right (**C1**). C2. The results of three independent experiments are presented in (C1) (*n* = 5). **D** Western blot analysis of TMPRSS11E, TFR1, IFN-γR2, phosphorylated STAT1, and iNOS in THP-1 differentiated M1 and M2 cells. **E** Immunofluorescence analysis of IFN-γR2 in M1- and M2-differentiated THP-1 cells. Representative fluorescence images. Scale bars, 50 μm (*n* = 5). **F** Western blot analysis of IFN-γR2 in the isolated endosome or plasma membrane from cultured cells. GAPDH in the cytosol was used as an internal controls.

markers confirmed the polarization of THP-1-derived macrophages (Supplementary Fig. 5). Western blot analysis revealed that during the THP-1-derived macrophage polarization process, TMPRSS11E expression was induced specifically in M1 macrophages and not in M2 macrophages. During the polarization of THP-1-derived macrophages to M1, the TMPRSS11E level gradually increased, and the TFR1 level in the cell lysate decreased (Fig. 7A, Supplementary Fig. 4D). Immunocytochemical staining also confirmed that TMPRSS11E was specifically expressed in M1 macrophages and that TFR1 was decreased, whereas in M2 macrophages, no TMPRSS11E was expressed, and TFR1 was increased (Fig. 7B). The cell-surface TFR1 level in individual cells is linearly related to holo-Tf uptake[22]. The uptake of Alexa-488-conjugated Tf into THP-1-derived M1 or M2 macrophages was measured. The mean fluorescence intensity (MFI) of Alexa-488-labeled Tf-treated cells was used as an indicator of fluorescently labeled ligand binding or uptake. M2 cells with relatively high surface TFR1 levels incorporated a large amount of Alexa-488-conjugated Tf, whereas

only modest incorporation was observed for M1 cells, whose cell surface TFR1 level was relatively low (Fig. 7C). Taken together, in M1 macrophages, significantly upregulated TMPRSS11E expression was also correlated with a reduction of cell surface TFR1- and TFR1-mediated iron uptake. Next, we wanted to study the effect of TMPRSS11E on the IFN-γR2 level after macrophage polarization. The western blot results revealed that compared with M2 macrophages, M1 macrophages presented lower TFR1 and higher IFN-γR2 levels; greater levels of IFN-γR2-dependent phosphorylated STAT1 and iNOS induction were also observed in M1 macrophages (Fig. 7D, Supplementary Fig. 4E). Moreover, M1 macrophages presented more IFN-γR2 on the cell surface (Fig. 7E, F, Supplementary Fig. 4F). These data suggest that in an inflammatory environment, M1 macrophages generate signals that favor IFN-γR2 accumulation on the cell surface, whereas an anti-inflammatory environment promotes IFN-γR2 internalization and induces M2 macrophage unresponsiveness to IFN-γ signaling.

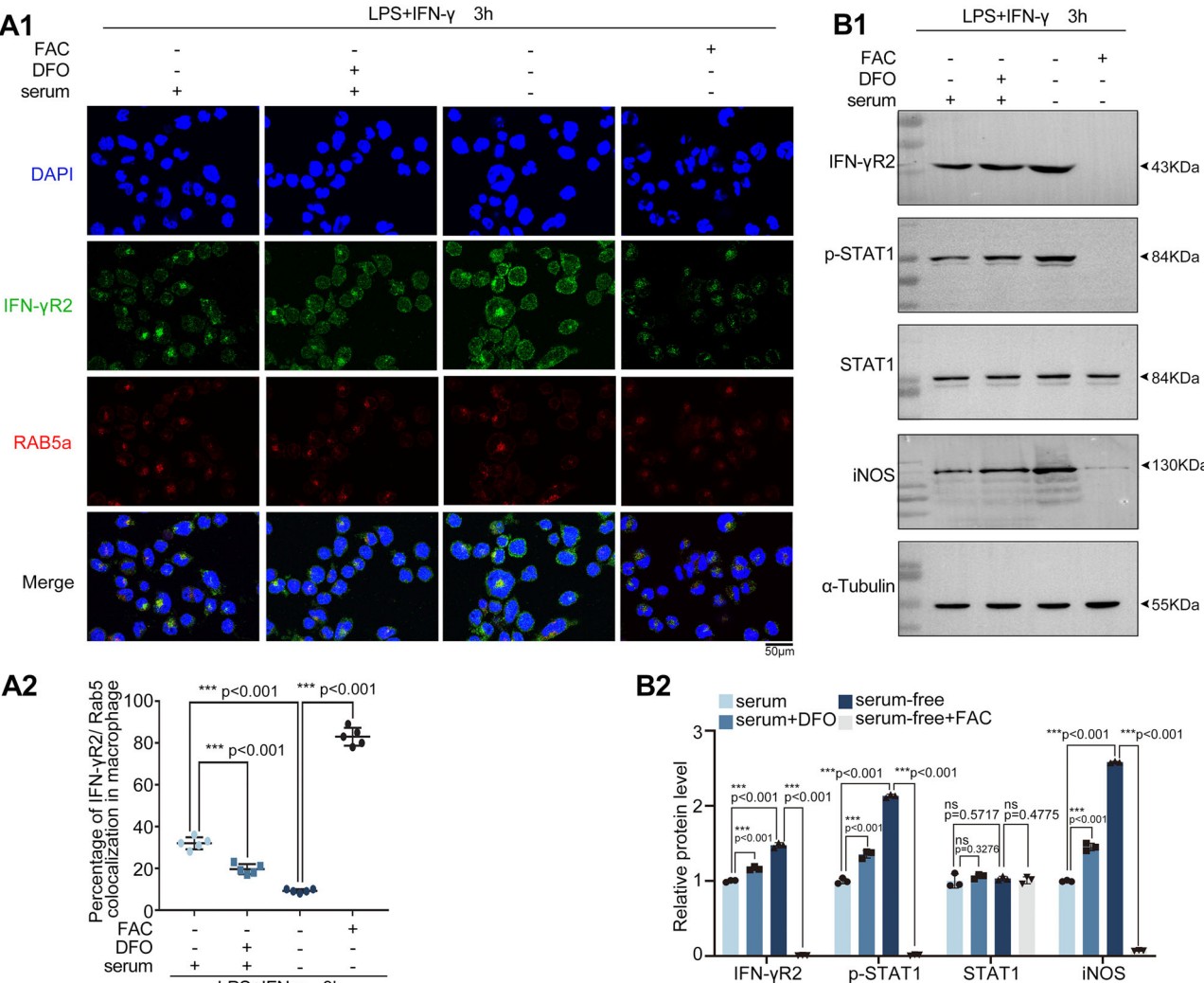

**Fig. 8 | Effect of iron on IFN-γR2 internalization and macrophage polarization. A** Double fluorescence staining for IFN-γR2 and Rab5a. Totally four experimental groups were prepared: cells were cultured in serum-free culture medium as the control group 1, in serum-containing complete culture medium as the control group 2. In parallel, to evaluate iron concentration on the IFN-γR2 distribution, complete medium supplemented with 10 μM deferoxamine (DFO) to remove free ferric iron (Fe³⁺) and decrease serum ferritin levels as group 3. Serum-free medium supplemented with 10 μM ammonium ferric citrate (FAC) to increase the iron concentration as group 4. THP-1-derived macrophages were cultured for 24 h in this medium and subsequently stimulated with LPS + IFN-γ for 3 h. Then, double fluorescence staining for IFN-γR2 and Rab5a was performed. A1. Representative fluorescence images. Scale bars, 50 μm. A2. The colocalization of internalized IFN-γR2 with Rab5a was calculated ($n = 5$). **B** Western blot analysis of the effect of iron on the activation of STAT1 during macrophage polarization. B1. Representative western blot analysis. B2. Quantitation of protein levels in (B1) ($n = 3$).

Several studies have demonstrated that the internalization of IFN-γR2 serves as a regulatory mechanism to modulate the strength and duration of IFN-γ signaling and prevent excessive inflammatory responses. IFN-γR2 receptor internalization is tightly linked to the fine-tuning of the JAK-STAT pathway[29,31].

IFN-γR2 internalization within T cells depends on iron uptake. Next, we evaluated the effects of iron uptake on the IFN-γR2 distribution during M1 macrophage polarization. Ferric ammonium citrate (FAC) and Deferoxamine (DFO) are used to modulate iron uptake. FAC is used extensively in the food industry as an additive and in medicine to treat iron-deficiency anemia in humans. In addition, FAC has become a standard source of iron in numerous biological studies. Increased cytosol labile iron $Fe^{2+}$ content was evident in cells treated with FAC. Opposite effects were observed for treatments with the iron-chelating agent DFO[32]. DFO is approved by FDA to treat iron overload. Deferoxamine binds to iron and removes it from the bloodstream. It functions by binding free ferric iron ($Fe^{3+}$) with high affinity, forming a stable complex that prevents iron from participating in cellular processes[26]. DFO has a significant association with a reduction in serum ferritin levels[33]. Here, THP-1 cells were activated toward the M1 phenotype in the presence of different iron conditions and the expression of IFN-γR2 and the internalization marker Rab5a was measured by immunofluorescence (Fig. 8A1). The colocalization of IFN-γR2 and Rab5a was examined to assess the effect of iron uptake on the endocytosis of IFN-γR2 (Fig. 8A2). Immunofluorescence revealed increased cell surface IFN-γR2 and decreased cytoplasmic IFN-γR2 when the iron supply decreased. When ferric ammonium citrate (FAC) was present, IFN-γR2 was detected almost exclusively in the cytosol and led to the most abundant colocalization between IFN-γR2 and Rab5a (Fig. 8A), which indicated that decreased IFN-γR2 at the membrane was accompanied by increased endocytosis. These data suggest that a deficiency in iron uptake increases the level of surface IFN-γR2 during M1 macrophage polarization. The western blot results revealed that when no iron was present, the IFN-γR2 level was the most abundant (Fig. 8B). The cellular response to IFN-γ responsiveness was also evaluated and confirmed, as p-STAT1 and iNOS expression was upregulated during M1 macrophage polarization. These results suggest that iron uptake by TFR1

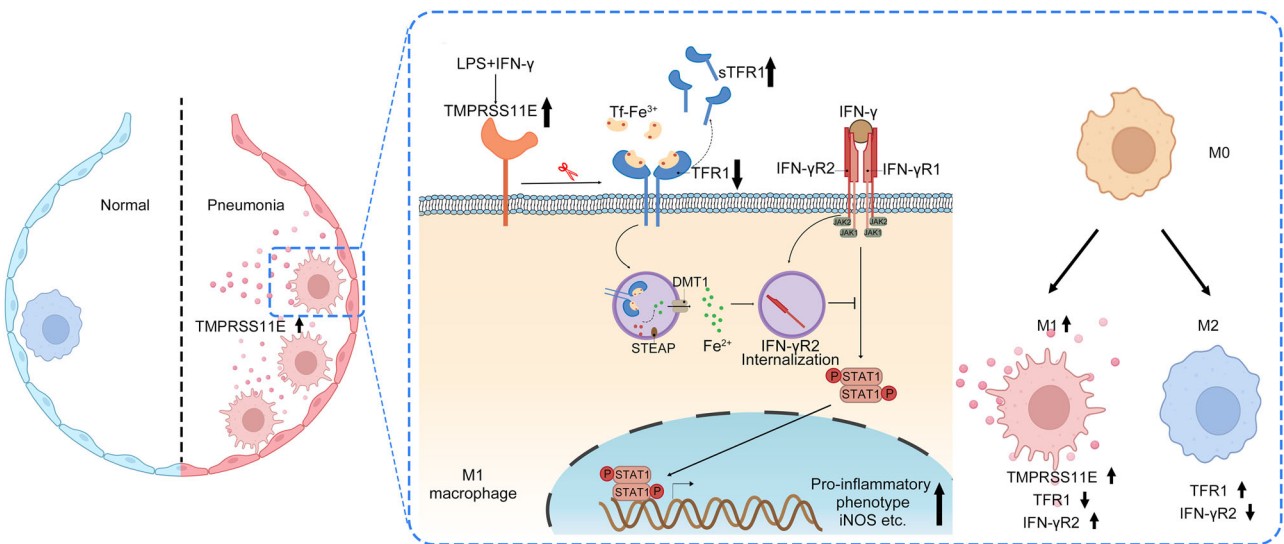

**Fig. 9 | The mechanism by which TMPRSS11E regulates the macrophage IFN-γ-STAT1 response pathway by cleaving surface TFR1 during inflammation.** In macrophages, LPS induced the expression of TMPRSS11E, which then cleaved the cell-surface transferrin receptor TFR1, releasing sTFR1. The increased sTFR1 may be a marker of inflammatory status. The cleavage of TFR1 by TMPRSS11E affects iron uptake and the internalization of IFN-γR2. The increased presence of IFN-γR2 on the cell membrane sensitizes IFN-γ signaling and promotes the macrophage pro-inflammatory M1 polarization. DMT1 divalent metal transporter 1, STEAP six-transmembrane epithelial antigen of prostate.

also influences IFN-γR2 endocytosis in macrophages and thus further influences macrophage polarization and the immune response.

## Discussion

In our previous research, TMPRSS11E expression was found to be induced in inflammatory macrophages. Interestingly, TMPRSS11E levels were also found to be increased in the tears of COVID-19 patients[34]. In the present study, we show, for the first time, that TFR1 can be cleaved by TMPRSS11E. Along with upregulated TMPRSS11E expression in M1 macrophages, increased sTFR1 levels were also found in the BALF of clinical patients with acute lung inflammation and in LPS-treated animal models. Our results are consistent with the previous report that sTFR1 levels are increased in the airways of patients with ARDS[13]. sTFR1 is also a potential biomarker for sepsis[35]. In addition, by analysis of patients from a tertiary rheumatology center, it was found that there was a positive correlation between sTFR1 levels with inflammatory markers[36]. Elevated sTFR1 also predicts poor prognosis in patients with COPD and may serve as a biomarker for severe pulmonary hypertension in COPD, identifying a distinct phenotype with systemic inflammation[37]. All these reports showed that sTFR1 production was associated with inflammation. Our findings first elucidate the mechanism of sTFR1 production in macrophages.

In addition, IFN-γ is a well-established and widely used stimulus for inducing M1 polarization in macrophages. During M1 macrophage polarization, TMPRSS11E was induced to cleave TFR1. TFR1 and IFN-γR2 are both cell-surface receptors whose trafficking involves clathrin-coated pit-mediated endocytosis. And treatment of cells with FAC resulted in increased in numbers and sizes of endosomes, opposite effects were observed for treatments with DFO[32]. Decreased TFR1-mediated iron uptake resulted in decreased IFN-γR2 internalization, and increased IFN-γR2 accumulation on the cell surface promoted the IFN-γ–STAT1 proinflammatory response. Therefore, our results reveal an immunoregulatory function of TMPRSS11E through affecting TFR1-mediated iron uptake in macrophages (Fig. 9). The identification of the association of iron uptake in IFN-γR2 internalization sheds new light on the fine-tuning of the IFN-γ/STAT1 pathway in macrophage cells.

Previously, TMPRSS6 was shown to cleave TFR1[38,39], but TMPRSS6 was shown to be expressed mainly in the liver[40]. Although TFR1 cleavage by TMPRSS6 has been reported, the physiological and pathological significance of TFR1 cleavage has not been investigated.

It was reported that alterations in the iron content of macrophages can significantly influence the modulation of innate immune responses[41]. For example, macrophages loaded with iron lose their ability to kill intracellular pathogens via IFN-γ-mediated pathways[42,43]. As a further consequence of intracellular iron depletion in macrophages, the proinflammatory immune effect is augmented. Elevated iron levels increase the proportion of macrophages with the M2 phenotype and negatively regulate the M1 proinflammatory LPS-induced response[44]. Iron reduces the M1 polarization of RAW264.7 macrophages[45]. When subjected to chronic iron overload, THP-1-derived macrophages polarize toward the M2 phenotype[46,47]. Our results are consistent with those observations and suggest that TMPRSS11E could decrease iron uptake by cleaving TFR1 and decreasing IFN-γR2 endocytosis in macrophages, thus promoting M1 macrophage polarization.

TFR1 is crucial for cellular iron uptake and plays a significant role in immunity. The TFR1 R22W mutation is associated with impaired immune function[48]. TFR1 R22W in a Turkish patient led to combined immunodeficiency and confirmed the iron-immunity axis. TFR2 is another receptor for Tf that is homologous to TFR1. TFR2 can also bind transferrin and deliver iron to cells. In mice, TFR2 deficiency polarizes macrophages toward a proinflammatory state that contributes to the development and progression of chronic inflammation in rheumatoid arthritis[49]. Compared with TFR2[+/+] macrophages, TFR2[-/-] macrophages presented increased expression of M1-like cytokines and prolonged STAT1 activation in response to IFN-γ stimulation. Our results showed that the consequences of TFR1 cleavage by TMPRSS11E were similar to those in TFR2[-/-] macrophages, suggesting that the observed effects are mediated via TFR2- or TFR1-controlled iron uptake.

It was reported that in the physical condition of lung tissue, TMPRSS11E expression was only detected in ionocytes[50]. In our previous studies, it was shown that under an inflammatory situation, TMPRSS11E was induced to be expressed primarily in macrophages, although expression of TMPRSS11E in epithelial cells was also induced[4]. As for TFR1, Public databases (the Human Protein Atlas) indicate that expression of TFR1 could be detected in most tissues. In lung tissue, more abundant TFR1 expression in alveolar macrophages was observed (https://www.proteinatlas.org/ENSG00000072274-TFRC/single+cell/lung). Therefore, under an inflammatory situation, sTFR1 processed by TMPRSS11E might primarily take place in macrophages. However, this needs more investigation in the future.

In conclusion, our results showed that TMPRSS11E cleaves TFR1 and releases sTFR1 in pro-inflammatory macrophages, indicating that increased sTFR1 might be a marker for inflammation situation. In addition, TMPRSS11E cleaves TFR1, decreases macrophage iron uptake, and affects IFN-γR2 internalization. Sensitization of IFN-γ signaling was accompanied by increased cell membrane IFN-γR2 location. Our findings suggest that macrophage TMPRSS11E exacerbates inflammation progression. These results reveal a previously unknown mechanism by which TMPRSS11E affects the inflammatory response and offer a new potential target for the treatment of pulmonary inflammation.

## Methods

### Cell culture, transfection and cell treatment
293 T cells (human embryonic kidney cells transformed with large T antigen), the human lung adenocarcinoma cell line A549 (commonly used as a model of malignant alveolar type II epithelial cells), and THP-1 cells (a human monocytic cell line) were purchased from American Type Culture Collection (ATCC; Manassas, VA, USA). THP-1 monocytes were differentiated into macrophages by incubation with phorbol 12-myristate 13-acetate (PMA, 100 ng/mL) for 12 h, followed by culture in RPMI medium. Macrophages were polarized into M1 macrophages by incubation with LPS (100 ng/mL) and IFN-γ (20 ng/mL) for 24 h. M2 macrophage polarization was induced by incubating macrophages with IL-4 (20 ng/mL) for 24 h.

### Generation of stable cell lines
TMPRSS11E and TMPRSS11E S372A mutant expression plasmids were constructed using the pLV3-CMV-MCS-Flag-Puro plasmid. To produce recombinant lentiviruses, the above vectors were cotransfected with packaging plasmids into 293 T cells to obtain lentiviral particles. To generate stable cell lines, THP-1 cells were infected with lentivirus that expresses TMPRSS11E or TMPRSS11E S372A. Stable cells were selected using 5 µg/ml puromycin.

### Plasmids and reagents
Plasmids, antibodies, and reagents used in this study are listed in Supplementary Tables 1 and 2. For the GFP-tagged TMPRSS11E construct pCMV-GFP-TMPRSS11E, it was designed to add the GFP tag at the N terminus of the full-length TMPRSS11E. Human TMPRSS11E has a short cytoplasmic region followed by a transmembrane region. The extracellular part of TMPRSS11E consists of an SEA domain followed by the C-terminal trypsin-like serine proteinase domain. The SEA domain is a conserved structural motif found in various membrane-bound serine proteases. It was suggested that the SEA domain more likely functions by orienting the active site cleft of TMPRSS11E toward plasma and/or extracellular spaces and away from the cell surface and/or the extracellular matrix. The C-terminal trypsin-like serine proteinase domain was a catalytically active form of the protease[51,52].

### Recombinant TMPRSS11E protein
The recombinant construct pET-32a-TMPRSS11E (AA 192-423) was transformed into *Escherichia coli* BL21 (DE3) to express the C-terminal serine proteinase domain, which represents the catalytically active form of TMPRSS11E[3]. Briefly, after induction with IPTG, the cells were harvested, lysed, and sonicated. The recombinant TMPRSS11E protein was purified by immobilized metal affinity chromatography (IMAC) using NI-NTA columns. The collected recombinant TMPRSS11E protein was subsequently dialyzed in PBS buffer, and endotoxins were removed.

### Coimmunoprecipitation
HEK293T cells were transfected with pCMV-TMPRSS11E-Flag and pCMV-V5-TFR1-myc, and total protein was extracted with RIPA lysis buffer. The lysates were then incubated with anti-Flag antibody or control rabbit IgG overnight, and protein A + G beads were subsequently used to capture the immune complexes. Then, western blotting was performed using an anti-V5 antibody.

### Liquid chromatography-tandem mass spectrometry (LC-MS/MS)
Immunoprecipitated protein complexes, including the bait TMPRSS11E and its corresponding negative control, were resolved by 10% SDS−PAGE followed by Coomassie Brilliant Blue staining. The resolved protein complexes were cut from the gel and subsequently subjected to in-gel digestion. The gel bands were first decolorized, followed by overnight trypsin digestion of the proteins at 37 °C. The digested peptides were extracted the next day, and the samples were vacuum-dried and resuspended in loading buffer. The peptides were analyzed using a nanoLC.2D (Eksigent Technologies) coupled to a TripleTOF 5600+ System (AB SCIEX, Concord, ON). The raw MS/MS data were first analyzed using ProteinPilot Software (version 4.5, SCIEX). The mgf files obtained from ProteinPilot were subsequently submitted to Protein Prospector (version 5.19.1, UCSF) and searched against the UniProt Human database (04/08/2019, 20,419/559,228 entries searched). To obtain proteins of interest, a secondary search comparison was performed to enable the summarization, identification, and comparison of all the search results. All the parameters were set as previously described[53].

### Animal models and hematoxylin and eosin (H&E) staining
Male C57BL/6 mice (aged 6–8 weeks) were purchased from Model Animal Research Center (MARC; Nanjing, China). The LPS-induced mouse model was generated as previously reported by intratracheal administration (2 mg/kg)[4]. After stimulation for the indicated times, the mice were euthanized, and lung tissues were obtained, fixed in formalin, embedded in paraffin, and sectioned. The tissue sections were then deparaffinized, rehydrated, and stained with hematoxylin and eosin. We have complied with all relevant ethical regulations for animal use.

### Western blot analysis
Western blot tests were repeated at least three times for each sample. Total protein from mouse lung tissue or cultured cells was extracted with RIPA buffer. Western blotting was conducted as previously described[4]. To ensure consistent results in Western blot experiments, the same batch of proteins for loading controls was used across different experiments. Quantification of Western blot signals was performed using ImageJ software. For each sample lane, the area around the protein band of interest and the α-Tubulin loading control was defined as the region of interest. After background subtraction, the integrated density of each band was obtained. The intensities of target proteins were normalized to α-Tubulin control to correct for loading variations. Data from a minimum of three independent experiments were included in the analysis. Quantification data are provided in the Supplementary Data 1. The graph representing the western blot quantification of the protein relative levels is also provided.

### Immunohistochemical (IHC) staining
Formalin-fixed paraffin-embedded (FFPE) mouse lung sections were deparaffinized and rehydrated. The sections were subsequently blocked and incubated with a primary antibody (TMPRSS11E antibody, CD68 antibody) overnight at 4 °C. Then, the sections were incubated with an HRP-conjugated secondary antibody followed by 3,3′-diaminobenzidine (DAB) staining, and hematoxylin was used for nuclear counterstaining. Staining results were classified as either positive or negative based on the presence of specific brown DAB staining in the target cells.

### Immunofluorescence (IF) assay
Immunofluorescence staining of the tissue was performed as previously reported[4]. FFPE mouse lung sections were deparaffinized and rehydrated. Antigens were subsequently retrieved, and the sections were incubated with primary antibodies overnight at 4 °C (TMPRSS11E antibody), followed by incubation with fluorescent dye-conjugated secondary antibodies for 1 h at room temperature. Nuclei were counterstained with DAPI. Double immunofluorescence staining of cultured cells was performed as previously reported[54]. Briefly, cells were washed with PBS and fixed in 4% paraformaldehyde, permeabilized with mild condition of 0.03% Triton X-100 for

20 minutes, which was optimized through preliminary experiments. After permeabilization cells were blocked with PBS containing 2% BSA for 1 h at room temperature. Then cells were stained with specific antibodies, including anti- TMPRSS11E, anti-IFN-γR2 and anti-Rab5a at 4 °C overnight, followed by species-matched fluorescent secondary antibodies and the nuclei were counterstained with DAPI. For colocalization analysis, images were acquired from at least five randomly selected fields per sample using a fluorescence microscope. The proportion of colocalization was calculated as the percentage of cells showing overlapping signals of the two fluorophores relative to the total number of cells within each field.

### Alexa Fluor® 488-labeled transferrin (Alexa-Transferrin) uptake assay

After cells were cooled to 4 °C on ice, they were incubated with 25 μg/mL Alexa-transferrin for 20 minutes (to allow ligand attachment to the cell surface and internalization) at 37 °C; subsequently, the cells were washed twice with precooled PBS and then fixed. Nuclei were counterstained with DAPI and observed under an Olympus laser confocal microscope. Fluorescence intensity of Alexa-transferrin (Tf-488) was quantified using ImageJ software. For each field, a region of interest (ROI) was defined around individual cells, and the mean fluorescence intensity was measured.

### ELISA for the quantitative measurement of soluble transferrin receptor 1 (sTFR1)

An antibody specific for TFR1 was precoated onto a microplate. Standards and samples (100 μl) were pipetted into wells and incubated for 2 h at 37 °C, and any TFR1 present was bound by the immobilized antibody. After the unbound substances were removed, a biotin-conjugated antibody specific for TFR1 was added to the wells, and the plates were then incubated for 1 h. After washing, avidin-conjugated horseradish peroxidase (HRP) was added to the wells, and the plates were subsequently incubated for 1 h. Following a wash to remove the unbound avidin-enzyme reagent, the substrate mixture was added, and the color was developed and measured at 450 nm. The results were calculated using a standard curve.

### Alveolar macrophage (AM) extraction from LPS-treated Sprague–Dawley (SD) rats

Male SD rats (aged 6–7 weeks) were purchased from the Model Animal Research Center (MARC; Nanjing, China). First, the rats were treated with LPS (2 mg/kg) by intranasal injection for the indicated times. Then, the rats were euthanized, and ice-cold PBS (0.6 ml) containing EDTA (10 nM) was infused into the lungs and withdrawn by tracheal cannulation twice. BALF samples were collected. For macrophage isolation, the collected BALF was centrifuged and suspended in DMEM containing 10% FBS for 2 h at 37 °C in plastic culture dishes. Nonadherent cells were removed, and the adherent macrophages were used as alveolar macrophages for subsequent experiments.

### Mouse peritoneal macrophage (PM) isolation

Peritoneal macrophages were obtained as previously reported. Briefly, 6–8-week-old C57BL/6 male mice (MARC, Nanjing, China) were intraperitoneally injected with 1 mL of sterile 3% thioglycollate. Four days after injection, the mice were euthanized, and the cells were harvested by lavage of the peritoneal cavity with 10 mL of DMEM. After adhesion to plastic culture dishes for 2 h, nonadherent cells were removed by washing three times with phosphate-buffered saline (PBS), and adherent macrophages were used for subsequent experiments.

### Patients and BALF collection

A total of 20 patients with acute lung inflammation and 10 control individuals were recruited for clinical analysis at Zhongda Hospital, Southeast University, China. Ten patients who were admitted for the evaluation of solitary pulmonary nodules (SPNs) without evidence of pulmonary infection composed the control group (Supplementary Table 3). BALF samples were collected following a standardized protocol by flushing the lungs with

saline and centrifuging the recovered material. The pelleted cells were suspended in DMEM and plated. After adhering to plastic culture dishes, the nonadherent cells were removed, and alveolar macrophages were used for subsequent analyses. All ethical regulations relevant to human research participants were followed.

### Flow cytometry

Differentiated THP-1 macrophages were incubated with anti-CD86 or anti-CD163 primary antibodies. Coralite 488-labeled goat anti-rabbit IgG was used as the secondary antibody. Flow cytometry was performed, and data analysis was performed with FlowJo software.

### Isolation of Membrane, Endosome

The Plasma Membrane Protein Isolation kit was used to isolate the membrane. Briefly, $3 \times 10^7$ cells were collected and added to buffer A, and then the cell suspension was placed on ice for 10 min. Then, gradient centrifugation was performed, and the plasma membrane or cytosol was obtained. The Endosome Isolation and Cell Fractionation kit was used to isolate endosomes. Briefly, $5 \times 10^7$ cells were collected and added to buffer A. Then the cell suspension was incubated on ice for 10 min. The filter cartridge filled with cell suspension was centrifuged. The supernatant was transferred to a fresh tube, and buffer B was added and incubated for 1 h. After centrifugation, the endosome pellet and cytosol supernatant were obtained.

### Statistics and reproducibility

The statistical analysis was performed using Student's $t$-test via GraphPad Prism 7. The data are presented as means ± standard deviations (SDs) of 3 independent experiments. $P < 0.05$ was considered statistically significant.

### Ethics approval and consent to participate

Informed consents were obtained from each individual. The study was approved by the Ethics Committee of the Zhongda Hospital. The experimental protocol for animals was approved by the Animal Experimentation Ethics Committee of Southeast University.

### Reporting summary

Further information on research design is available in the Nature Portfolio Reporting Summary linked to this article.

## Data availability

The source data behind the graphs in the paper can be found in Supplementary Data 1. Unprocessed original images of Western blots are provided in Supplementary Fig. 6.

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

## Acknowledgements

This work was supported by a grant from the Chinese National Natural Science Foundation (31070706). We are grateful to the Doctors of the Department of Respiratory of Zhongda Hospital, Southeast University, for their assistance with patients' BALF collection.

## Author contributions

Ting Wang: performed experiments, analyzed the data, and compiled figures. Zhenfa Chen and Nannan Wang: performed MS experiment. Yiwei Jiang, Wei Zhang, and Jie Ding: performed discussions. Xihua Wang and Ling Liu: provided patient samples. Lei Fang: provided MS analysis. Zichun Hua: provided suggestions and corrections. Shufeng Li: designed experiments and analyzed the data, wrote the manuscript. All authors reviewed the manuscript.

## Competing interests
The authors declare no competing interests.
