## [Transparent Peer Review file · Communications Biology]

TMPRSS11E-mediated TFR1 cleavage influences IFN- γ R2 internalization and the macrophage innate response

Corresponding Author: Professor Shufeng Li

Version 0:

Reviewer comments:

Reviewer #1

(Remarks to the Author)

The authors expand on their 2023 paper, which described the upregulation of the serine protease TMPRSS11E in pulmonary inflammation. In this manuscript, they propose a mechanism of action for TMPRSS11E in macrophages using in vitro and in vivo models as well as patient samples. They highlight its ability to cleave TFR1, regulate IFN- γ R2 internalization and iron uptake, and thereby influence macrophage differentiation/polarization.

While the manuscript presents a comprehensive study and the proposed mechanism demonstrates novelty, it has several significant shortcomings and cannot be accepted in its current form.

Major Comments:

1. Clarity of Aim: The abstract, intro and experiments predominantly focus on pulmonary inflammation, but the conclusion of manuscript emphasizes sepsis treatment and diagnosis as the primary application. This inconsistency needs to be resolved. Either the entire manuscript should align with the focus on sepsis, or the experimental setup and conclusions should reflect pulmonary inflammation as the primary aim.
2. Insufficient Discussion: The discussion lacks depth and should incorporate more precise references to relevant literature. In some cases, data are partially discussed within the results sections, leading to a fragmented narrative. These elements should be unified and expanded in the discussion.
3. Missing References: The section discussing IFN- γ receptors in macrophages (Rows 219–229) is entirely devoid of references. Proper citations are required to support the statements made.
4. Use of Animal Models: The rationale for employing both rat and mouse models to demonstrate similar findings is unclear. Furthermore, the use of peritoneal macrophages in the mouse model warrants explanation.
5. Statistical Analysis: The manuscript lacks detailed statistical descriptions. For instance: The number of repetitions for Western blot analyses is not stated, and densitometry is not included. The immunofluorescence analyses mention three independent repetitions, but five data points are depicted. Were additional values from the same experiments analysed?
6. Figure Legends: Some figure legends (e.g., Figure 2) are overly detailed and confusing. Less critical information, such as incubation times, should be moved to the methodological section.
7. Carmostat Inhibitor Effect: Figure 5B suggests that the Carmostat inhibitor enhances TFR1 and TMPRSS11E levels. This observation should be addressed and explained.
8. Rab5a Data Interpretation: The role of Rab5a in endocytosis/phagocytosis should be explained earlier to clarify its significance in Figure 6. Additionally, the interpretation in Rows 233–236 should be revised for better clarity.
9. Relevance of IFN- γ Treatment: The authors should discuss the relevance of using IFN- γ as a polarization agent alongside the evaluation of IFN- γ R2 expression in M1 macrophages.
10. Iron Concentrations: In the results section (Row 280–281), the authors refer to “different iron concentrations,” but the data only include two sources of iron, each tested at a single concentration. This phrasing should be corrected or clarified.
11. Western Blot Details: Clarify whether STAT phosphorylation was detected simultaneously with iNOS in the Western blot analysis. The methodology for Western blotting requires more comprehensive details.

Minor Comments:

1. Role in Other Cell Types: The authors should discuss whether TMPRSS11E might also play a role in other cell types beyond macrophages.
2. Typographical error in Row 138: “aa” should be corrected.
3. The terms in vitro and in vivo should be italicized throughout the manuscript.
4. In Figure 2, concentrations or amounts of the enzyme should be explicitly stated rather than using symbols (+/+/+/+).
5. In Figure 3, it is unclear which mutant corresponds to which label. This should be clarified in the figure or legend.

6. The legend for Figure 4 should be reviewed and revised for accuracy and clarity.

Reviewer #2

(Remarks to the Author)

The manuscript describes the activity of serine protease TMPRSS11E in macrophages. The authors show that the protease interacts with the transferrin receptor and catalyzes its proteolysis, leading to the release of the soluble form of the receptor into the extracellular environment. Infection leads to increased expression of the protease, and consequently, enhanced proteolysis of the transferrin receptor. A deficiency of the receptor and iron uptake result in inhibited internalization of the IFN- γ R2 receptor, which in turn leads to its hyperactivation, aggravating pro-inflammatory signaling.

The study was performed on cell lines, but the key findings were verified in pulmonary macrophages of pneumonia patients and primary murine macrophages.

The topic is interesting and novel, as the TMPRSS11E has only recently been identified in macrophages by the authors. Similarly, the role of transferrin receptor shedding remains poorly understood. The authors propose an existence of a pathway involved in macrophage differentiation toward the M1 pro-inflammatory phenotype, which is based on reduced iron uptake and stabilization of IFN- γ R2 on the cell surface.

The experimental design is well thought out, the results are concisely described, and the figures appear convincing.

However, some important details are missing thus hindering the positive assessment of the results.

The following issues need to be clarified:

1. Immunofluorescence data:

Lines 231-232, the author writes: Immunofluorescence analysis revealed markedly higher levels of membrane IFN- γ R2 accumulation. – Fig. 6A.

Which membrane are they referring to? Is this the plasma membrane, as the text suggests? Enhanced staining of the cell perimeter suggests the plasmamembrane.

The authors are cautious in their description, but the meaning of their argument is clear and points to the plasma membrane surface. I assume that cells were permeabilized to detect Rab5, although this is not indicated in Methods. If that is the case, we cannot be certain which membrane or surface is being visualized without further analysis.

Furthermore, in lines 284-285, the authors write about Fig. 8: Immunofluorescence revealed increased cell surface IFN- γ R2 and decreased cytoplasmic IFN- γ R2 when the iron supply decreased. The same set of proteins, as in Fig. 6, is shown here, and the “surface level of the receptor” is clearly stated.

Also, lines 272-273 say: Moreover, M1 macrophages presented more IFN- γ R2 on the cell surface (Fig. 7E).

In all these cases important details about the procedure of cell staining are missing and the statement of surface level of IFN- γ R2 seems premature.

Other issues:

1. Lines 122-124. The authors write: The cell surface TFR1 level in the mutant TMPRSS11E-transfected group was significantly greater than that in the wild-type TMPRSS11E-transfected group, reflecting the impaired cleavage of mutated TMPRSS11E.

In lines 134-135, they write: The surface TFR1 level in the inhibitor group was similar to that in the empty control plasmid pCMV-GFP-transfected group.

It is not clear which data these conclusions are based on.

2. The TMPRSS11E construct with the GFP tag, shown in Figure 2C, is not sufficiently described. The Methods section should include basic information about it, such as the role of the SEA motif, etc.

3. It is worth to clearly state in the text that the “TMPRSS11E recombinant protein” is in fact its catalytically active fragment, and a reference should be made to the previous study of the authors in Carcinogenesis, where the construct is described in detail.

4. The legend to Fig. 4A, B, C, D: what does “IHC staining” refer to? Fig. 4B1, there is a similar simple statement about “Immunofluorescence staining”. Please, indicate which proteins are visualized in these images.

5. Lines 178-179, the author writes: Histological analysis revealed neutrophil infiltration in the lung tissues from mice in the LPS. What data are they referring to?

6. The description of the approaches used to modulate iron uptake is not sufficient. DFO is not identified as an iron chelator.

7. The legend of Fig. 8, lines 696-698, suggests that cells preincubated with ammonium ferric citrate were stimulated with LPS in serum-free medium, while cells treated with DFO - in medium containing serum. This is an important difference, since serum contains LPS-binding protein (LBP) which facilitates stimulation of cell with this endotoxin.

8. What was the chemotype of LPS used in the studies? In Table S2 “lipopolysaccharides” are mentioned, but no further detail is provided.

9. In Fig. 9, abbreviations such as DMT1 and STEAP appear but are not explained in the text.

10. Fig. s1B – lanes of the gel are not described, making the Figure difficult to interpret.

Version 1:

Reviewer comments:

Reviewer #1

(Remarks to the Author)

The authors have addressed most of the previous comments and have improved the overall quality of the manuscript. I appreciate the inclusion of the number of repetitions and the enhanced discussion.

To facilitate the review process, I recommend that, in future revisions, the authors indicate the location of the newly added or revised text (e.g., specific line numbers or sections in the manuscript) where reviewer comments have been addressed. This would be more effective than copying the revised text solely into the response letter.

One main concern that remains is the densitometric evaluation. Although the authors stated in their responses that densitometric analysis was performed to confirm the robustness of the data and that these analyses were incorporated into the new figures, I was unable to locate them. Furthermore, the methodology section and figure legends do not include any explanation or mention of densitometry. Please clarify this point.

Additionally, please briefly incorporate into the manuscript text a rationale for the use of both mouse and rat models that you answered to Rev1 to avoid potential confusion.

Finally, the manuscript should be carefully revised for grammar and typographical errors. For example:

- Line 328: “sTfR1” – please check context and formatting.
- Lines 376, 748, 795: incorrect or missing spaces.
- Line 715: missing word – possibly “analysis,” “blot,” or “blotting.”

Reviewer #2

(Remarks to the Author)

I maintain my positive opinion regarding the topic, scope of the study, and the interesting results. The Authors successfully addressed some of my concerns and provided source immunoblots. I have some reservations about the way the data are presented, which in turn affects the overall clarity of the text.

For example, in response to my previous comment 1, the authors indicate in the letter which gel lanes should be compared in Fig. 2. Unfortunately, this information is still missing from the manuscript itself. Furthermore, neither the figure, its legend nor Methods specify how the individual proteins were detected in immunoblotting analysis. Among others, the information on the detection of GFP-TMPRSS11E with an antibody specific against the protease itself, not the tag (GFP), can facilitate the analysis of the immunoblots. The term 'western' is used to refer to immunoblotting analyses. Similarly, in the case of immunocytochemistry staining and immunofluorescent detection, it is not stated which antibodies were used for the analysis. To keep the Figure legends concise, these details can be included in the respective sections of Methods with reference do Supplemental list of antibodies. At present, all the data on antibodies used in immunoblotting are included in the file “Related WB”, and there is also a list of antibodies found in the Supplement; however, it does not seem sufficient, as it does not specify which antibodies were used for which type of analysis (WB, IF, ICH). Regarding my previous remark 6 – the explanation of the methodological approach is provided in the letter; however, the description in the manuscript remains brief, and as a result, unclear. Incorporating these revisions should improve the readability of the text.

Version 2:

Reviewer comments:

Reviewer #1

(Remarks to the Author)

The authors have addressed the previous comments and improved the overall quality of the manuscript. In my view, the manuscript is now suitable for publication in Communications Biology.

Dear Referees,

We are very grateful to you for your critical and suggestive comments on our manuscript. The questions raised by the referees are responded as follows, and the amendments were highlighted in red in the revised manuscript.

Reviewers' comments:

Reviewer #1 (Remarks to the Author):

The authors expand on their 2023 paper, which described the upregulation of the serine protease TMPRSS11E in pulmonary inflammation. In this manuscript, they propose a mechanism of action for TMPRSS11E in macrophages using in vitro and in vivo models as well as patient samples. They highlight its ability to cleave TFR1, regulate IFN- γ R2 internalization and iron uptake, and thereby influence macrophage differentiation/polarization. While the manuscript presents a comprehensive study and the proposed mechanism demonstrates novelty, it has several significant shortcomings and cannot be accepted in its current form.

Major Comments:

1. Clarity of Aim: The abstract, intro and experiments predominantly focus on pulmonary inflammation, but the conclusion of manuscript emphasizes sepsis treatment and diagnosis as the primary application. This inconsistency needs to be resolved. Either the entire manuscript should align with the focus on sepsis, or the experimental setup and conclusions should reflect pulmonary inflammation as the primary aim.

Answer 1: We are very grateful to your kind, careful review and comments. According to your suggestion, we have carefully revised the manuscript, and focus on the pulmonary inflammation as the primary aim in the updated manuscript.

2. Insufficient Discussion: The discussion lacks depth and should incorporate more precise references to relevant literature. In some cases, data are partially discussed within the results sections, leading to a fragmented narrative. These elements should be unified and expanded in the discussion.

Answer 2: Thank you very much for your valuable comment. According to your suggestion, we have thoroughly revised the Discussion section. Specifically, we have deleted the discussion fragment in the results sections and incorporated more precise references in the discussion section in the revised manuscript.

3. Missing References: The section discussing IFN- γ receptors in macrophages (Rows 219–229) is entirely devoid of references. Proper citations are required to support the statements made.

Answer 3: Thank you very much for your suggestion. Now we have provided the references for the section discussing IFN- γ receptors in macrophages (Rows 219–229)[*Reference 1-2*].

4. Use of Animal Models: The rationale for employing both rat and mouse models to demonstrate similar findings is unclear. Furthermore, the use of peritoneal macrophages in the mouse model warrants explanation.

Answer 4: Thank you for your valuable comment. In vivo lung inflammation models are the best tool to study the biological role of a given pathogen or LPS. In practice, both LPS lung inflammation rat model and mice model were widely used [Reference 3, 4]. In vitro models of lung inflammation usually consist of stimulation of isolated pulmonary cells with respective pathogens or LPS. In practice, isolation of alveolar macrophage from mice is a relatively simple procedure that only requires performing a bronchoalveolar lavage in Mice. For low yield of mice bronchoalveolar lavage, which almost invariably requires a large number of animals to obtain sufficient numbers of cells. While BALF from Rats could provide much more alveolar macrophages [Reference5,6].

In addition, peritoneal macrophages are also widely used for studies of macrophage inflammation [Reference7-11]. Primary murine peritoneal macrophages and macrophage cell lines were used to support that in all these macrophage cells, TMPRSS11E could be induced.

5. Statistical Analysis: The manuscript lacks detailed statistical descriptions. For instance: The number of repetitions for Western blot analyses is not stated, and densitometry is not included. The immunofluorescence analyses mention three independent repetitions, but five data points are depicted. Were additional values from the same experiments analysed?

Answer 5: Thank you very much for your careful review. Western blot analyses were independently repeated at least three times, and the number of biological replicates has now been indicated in the revised figure legends. Densitometric analysis was performed using ImageJ software to quantify protein band intensities. In the revised figures, densitometry analysis for Western blot were added. For the immunofluorescence analyses, repeating the whole experiment independently were performed three times. In experimental design, duplicate wells for each condition were performed in the first and second times of experiments, and in the third time, one well was performed for one condition—total of five data points were obtained for each condition. We have now provided this information in the revised figure legend.

6. Figure Legends: Some figure legends (e.g., Figure 2) are overly detailed and confusing. Less critical information, such as incubation times, should be moved to the methodological section.

Answer 6: Thank you for your valuable suggestion. As you suggested, legend for Figure 2 were revised now.

7. Carmostat Inhibitor Effect: Figure 5B suggests that the Carmostat inhibitor enhances TFR1 and TMPRSS11E levels. This observation should be addressed and explained.

Answer 7: Thank you very much for your suggestion. Carmostat functions as a serine protease inhibitor that has been used in Japan to treat chronic pancreatitis. [Reference12], Carmostat is also used as a protease inhibitor to prevent unwanted drug degradation, improving oral bioavailability [Reference13]. Carmostat also efficiently inhibits the TMPRSS2 activity, eventually blocking the S protein cleavage and

consequently blocks SARS-CoV2 infection [Reference14]. Camostat was also reported to inhibit some protein degradation [Reference15]. Here it was used to inhibit the proteolytic activity of TMPRSS11E. Camostat inhibited the autocleavage of TMPRSS11E and might also inhibit other endogenous serine protease, as a result of this inhibition, TMPRSS11E protein lose its activity. The levels of full-length TMPRSS11E are increased, and inactivated TMPRSS11E could not cut TFR1. We have now revised the manuscript as you suggested.

8. Rab5a Data Interpretation: The role of Rab5a in endocytosis/phagocytosis should be explained earlier to clarify its significance in Figure 6. Additionally, the interpretation in Rows 233–236 should be revised for better clarity.

Answer 8: Thank you for your valuable comment. According to your advice, we have now included a brief explanation of Rab5a's function earlier in the Results section to improve clarity in the revised manuscript. Additionally, we have revised the interpretation in Rows 233–236 for better clarity.

Rab5a is a member of the Rab family of small GTPases and plays a crucial role in endocytosis. Rab5a is a marker of early sorting endosomes and used to indicate endocytic trafficking and vesicle maturation[Reference16]. Increased expression of Rab5a correlates with active endocytosis. Rab5 was associated with the internalization of receptors and signaling pathways[Reference17]. Endocytosis of receptors has been considered as a mechanism of signal attenuation. The receptors are sorted through the early endosome. Most of the receptor enter into the late endosomes and then degrade after reaching the lysosomes.

9. Relevance of IFN- γ Treatment: The authors should discuss the relevance of using IFN- γ as a polarization agent alongside the evaluation of IFN- γ R2 expression in M1 macrophages.

Answer 9: Thank you very much for your comment. As you suggested, we revised the manuscript.

IFN- γ is a well-established and widely used stimulus for inducing M1 polarization in macrophages. It activates classical pro-inflammatory pathways through its interaction with the IFN- γ receptor complex, which includes IFN- γ R1 and IFN- γ R2 subunits. Upon binding of IFN- γ to its receptor complex, signal transduction is initiated through the JAK-STAT pathway, leading to the recruitment and phosphorylation of STAT1, which then translocates to the nucleus to drive transcription of pro-inflammatory genes[Reference18].

Notably, abnormalities in IFN- γ R1 and IFN- γ R2 expression are closely associated with Mendelian susceptibility to mycobacterial disease (MSMD), a syndrome characterized by localized or disseminated infections caused by atypical mycobacteria [Reference19]. Given the critical roles of IFN γ R2 in immunity and disease, understanding the mechanisms controlling IFN- γ R2 membrane location in immune cells such as macrophages has important biological significance and clinical implications. Several studies have demonstrated that IFN- γ R2 receptor internalization

is tightly linked to the fine-tuning of the JAK-STAT pathway [Reference20- 21]. The internalization of IFN- γ R2 serves as a regulatory mechanism to modulate the strength and duration of IFN- γ signaling and prevent excessive inflammatory responses.

10. Iron Concentrations: In the results section (Row 280–281), the authors refer to “different iron concentrations,” but the data only include two sources of iron, each tested at a single concentration. This phrasing should be corrected or clarified.

Answer 10: Thank you very much for your careful comment. As you suggested, we have revised the Figure legend for Fig.8. Please refer to the new Fig. 8 in the current resubmitted revision.

Serum free medium represents a simple and convenient way to completely deplete iron provide from serum. Complete medium containing serum stand for the normal iron concentration culture. In parallel, Serum containing complete medium supplemented with DFO represents the condition that the iron in the serum was partially chelated. Serum free medium supplemented with FAC stand for the high iron concentration with addition of iron source FAC. Therefore, serum-free medium, complete medium with DFO, complete medium, serum-free medium with FAC stand for four different iron conditions [References22].

11. Western Blot Details: Clarify whether STAT phosphorylation was detected simultaneously with iNOS in the Western blot analysis. The methodology for Western blotting requires more comprehensive details.

Answer 11: Thank you for your good comment. As you suggested, we have revised the manuscript. STAT phosphorylation and iNOS were detected on separate membranes. To maintain consistency in Western blot (WB) experiments, the same batch of proteins were used for performing WB. Housekeeping proteins GAPDH were used for WB normalization.

Minor Comments:

1. Role in Other Cell Types: The authors should discuss whether TMPRSS11E might also play a role in other cell types beyond macrophages.

Answer 1: Thank you for your valuable comment. We have now added a brief discussion of TMPRSS11E’s potential role in other cell types in the revised Discussion section.

It was reported that in physical condition, TMPRSS11E expression in lung was very low and TMPRSS11E was primarily expressed in ionocytes [Reference 23]. In our previous studies, it was shown that under inflammation situation, TMPRSS11E was induced to express primarily in macrophage. And in epithelial cells TMPRSS11E was also induced to express [Reference 24]. As for TFR1, Public databases (the Human Protein Atlas) indicate that expression of TFR1 could be detected in most tissues. In addition, in lung tissue, highly abundant TFR1 expression in alveolar macrophages was observed compared to other pulmonary cell types based on single-cell RNA sequencing data. The figure of the cell-type-specific expression pattern of TFR1 in the

lung (<https://www.proteinatlas.org/ENSG00000072274-TFRC/single+cell/lung>) was shown here. Therefore, under inflammatory situation, sTFR1 processed by TMPRSS11E might primarily in macrophages. In epithelial might also contribute the sTFR1 release, however, this need more investigation in future.

2. Typographical error in Row 138: “aa” should be corrected.

Answer 2: Thank you for your correction. According to your advice, the term “aa” abbreviation for amino acids in Row 138 was revised in the current revised manuscript.

3. The terms *in vitro* and *in vivo* should be italicized throughout the manuscript.

Answer 3: Thank you for your careful review. The terms *in vitro* and *in vivo* have been italicized throughout the manuscript as recommended.

4. In Figure 2, concentrations or amounts of the enzyme should be explicitly stated rather than using symbols (+/+/+/+).

Answer 4: Thank you for your suggestion. We have revised Figure 2 to indicate the enzyme amounts.

5. In Figure 3, it is unclear which mutant corresponds to which label. This should be clarified in the figure or legend.

Answer 5: Thank you for your good suggestion. We have revised the figure and figure legend to include the detailed information of each mutant for clarity.

6. The legend for Figure 4 should be reviewed and revised for accuracy and clarity.

Answer 6: Thank you very much for your valuable suggestion. We have carefully revised the legend for Figure 4 to improve its accuracy and clarity.

References:

1. Regis G, et al. Iron regulates T-lymphocyte sensitivity to the IFN-gamma/STAT1 signaling pathway *in vitro* and *in vivo*. *Blood* 105:3214-3221 (2005).
2. Ivashkiv LB. IFN-gamma: signalling, epigenetics and roles in immunity, metabolism, disease and cancer immunotherapy. *Nat Rev Immunol.* 18:545-558

(2018).

3. Hui J, Aulakh GK, Unniappan S, Singh B. Loss of Nucleobindin-2/Nesfatin-1 increases lipopolysaccharide-induced murine acute lung inflammation. *Cell Tissue Res.* 385(1):87-103 (2021).
4. Guo Y, Liu Y, Zhao S, Xu W, Li Y, Zhao P, Wang D, Cheng H, Ke Y, Zhang X. Oxidative stress-induced FABP5 S-glutathionylation protects against acute lung injury by suppressing inflammation in macrophages. *Nat Commun.* 12(1):7094 (2021).
5. Patoine D, Bouchard K, Lemay AM, Bissonnette EY, Lauzon-Joset JF. Specificity of CD200/CD200R pathway in LPS-induced lung inflammation. *Front Immunol.* 13:1092126 (2022).
6. Liu Y, Zhou J, Luo Y, Li J, Shang L, Zhou F, Yang S. Honokiol alleviates LPS-induced acute lung injury by inhibiting NLRP3 inflammasome-mediated pyroptosis via Nrf2 activation in vitro and in vivo. *Chin Med* 16:127 (2021).
7. Du J, Wang G, Luo H, Liu N, Xie J. JNK-IN-8 treatment alleviates lipopolysaccharide-induced acute lung injury via suppression of inflammation and oxidative stress regulated by JNK/NF- κ B signaling. *Mol Med Rep.* 23(2):150(2021).
8. Luo XQ, Duan JX, Yang HH, Zhang CY, Sun CC, Guan XX, Xiong JB, Zu C, Tao JH, Zhou Y, Guan CX. Epoxyeicosatrienoic acids inhibit the activation of NLRP3 inflammasome in murine macrophages. *J Cell Physiol.* 235(12):9910-9921(2020).
9. Zhong WJ, Duan JX, Liu T, Yang HH, Guan XX, Zhang CY, Yang JT, Xiong JB, Zhou Y, Guan CX, Li Q. Activation of NLRP3 inflammasome up-regulates TREM-1 expression in murine macrophages via HMGB1 and IL-18. *Int Immunopharmacol.* 89(Pt A):107045 (2020).
10. Hsu CG, Fazal F, Rahman A, Berk BC, Yan C. Phosphodiesterase 10A Is a Key Mediator of Lung Inflammation. *J Immunol.* 206(12):3010-3020 (2021).
11. Shi H, Wang Y, Li X, Zhan X, Tang M, Fina M, Su L, Pratt D, Bu CH, Hildebrand S, Lyon S, Scott L, Quan J, Sun Q, Russell J, Arnett S, Jurek P, Chen D, Kravchenko VV, Mathison JC, Moresco EM, Monson NL, Ulevitch RJ, Beutler B. NLRP3 activation and mitosis are mutually exclusive events coordinated by NEK7, a new inflammasome component. *Nat Immunol.* 17(3):250-8(2016).
12. Yamaya M, Shimotai Y, Hatachi Y, Lusamba Kalonji N, Tando Y, Kitajima Y, Matsuo K, Kubo H, Nagatomi R, Hongo S, Homma M, Nishimura H. The serine protease inhibitor camostat inhibits influenza virus replication and cytokine production in primary cultures of human tracheal epithelial cells. *Pulm Pharmacol Ther* 33: 66-74 (2015).
13. Taoxing Peng, Xinyue Shaob, Li Long, Han Liu, Wenqin Song, Jiazhen Hou, Haijun Zhong, Yang Ding, Yongzhuo Huang. Rational design of oral delivery nanosystems for hypoglycemic peptides. *NanoToday.*53:102031(2023)
14. Shrimp JH, Kales SC, Sanderson PE, Simeonov A, Shen M, Hall MD. An Enzymatic TMPRSS2 Assay for Assessment of Clinical Candidates and Discovery of Inhibitors as Potential Treatment of COVID-19. *ACS Pharmacol Transl Sci.*3(5):997-1007(2020).
15. Inuzuka T, Sato S, Baba H, Miyatake T. Degradation of myelin basic protein in myelin by protease in cerebrospinal fluid and effects of protease inhibitors.

Neurochem Res. 11(10):1407-17(1986).

16. Zhang X, Chen C, Ling C, Luo S, Xiong Z, Liu X, Liao C, Xie P, Liu Y, Zhang L, Chen Z, Liu Z, Tang J. EGFR tyrosine kinase activity and Rab GTPases coordinate EGFR trafficking to regulate macrophage activation in sepsis. *Cell Death Dis.* 13(11):934(2022).

17. Chen F, Zhang D, Cheng L, Zhao D, Ye H, Zheng S, Yang Q, Han B, Wang R, Li J, Chen S. Xiaowugui decoction alleviates experimental rheumatoid arthritis by suppressing Rab5a-mediated TLR4 internalization in macrophages. *Phytomedicine.* 132:155762 (2024).

18. Xu X, Xu J, Wu J, Hu Y, Han Y, Gu Y, Zhao K, Zhang Q, Liu X, Liu J, Liu B, Cao X. Phosphorylation-Mediated IFN- γ R2 Membrane Translocation Is Required to Activate Macrophage Innate Response. *Cell.* 175(5):1336-1351.e17(2018).

19. Sharma, V.K., Pai, G., Deswarte, C. et al. Disseminated Mycobacterium avium Complex Infection in a Child with Partial Dominant Interferon Gamma Receptor 1 Deficiency in India. *J Clin Immunol* 35:459–462 (2015).

20. Rosenzweig SD, Schwartz OM, Brown MR, Leto TL, Holland SM. Characterization of a dipeptide motif regulating IFN-gamma receptor 2 plasma membrane accumulation and IFN-gamma responsiveness. *J Immunol* 173: 3991-3999 (2004).

21. Boselli D, Ragimbeau J, Orlando L, Cappello P, Capello M, Ambrogio C, Chiarle R, Marsili G, Battistini A, Giovarelli M, Pellegrini S, Novelli F. Expression of IFN γ R2 mutated in a dileucine internalization motif reinstates IFN γ signaling and apoptosis in human T lymphocytes. *Immunol Lett* 134:17-25 (2010).

22. Regis G, Bosticardo M, Conti L, De Angelis S, Boselli D, Tomaino B, Bernabei P, Giovarelli M, Novelli F. Iron regulates T-lymphocyte sensitivity to the IFN-gamma/STAT1 signaling pathway in vitro and in vivo. *Blood* 105: 3214-3221 (2005).

23. Hoffmann M, Hofmann-Winkler H, Smith JC, Krüger N, Arora P, Sørensen LK, Sjøgaard OS, Hasselstrøm JB, Winkler M, Hempel T, Raich L, Olsson S, Danov O, Jonigk D, Yamazoe T, Yamatsuta K, Mizuno H, Ludwig S, Noé F, Kjolby M, Braun A, Sheltzer JM, Pöhlmann S. Camostat mesylate inhibits SARS-CoV-2 activation by TMPRSS2-related proteases and its metabolite GBPA exerts antiviral activity. *EBioMedicine.* 65:103255 (2021).

24. Zhang W, Chen Z, Wang T, Wang X, Liu L, Huang W, Li S. Increased Expression of TMPRSS11E Is Involved in LPS- or Poly(I:C)-mediated Inflammation. *Am J Respir Cell Mol Biol.* 68: 406-416 (2023).

Reviewer #2 (Remarks to the Author):

The manuscript describes the activity of serine protease TMPRSS11E in macrophages. The authors show that the protease interacts with the transferrin receptor and catalyzes its proteolysis, leading to the release of the soluble form of the receptor into the

extracellular environment. Infection leads to increased expression of the protease, and consequently, enhanced proteolysis of the transferrin receptor. A deficiency of the receptor and iron uptake result in inhibited internalization of the IFN- γ R2 receptor, which in turn leads to its hyperactivation, aggravating pro-inflammatory signaling.

The study was performed on cell lines, but the key findings were verified in pulmonary macrophages of pneumonia patients and primary murine macrophages.

The topic is interesting and novel, as the TMPRSS11E has only recently been identified in macrophages by the authors. Similarly, the role of transferrin receptor shedding remains poorly understood. The authors propose an existence of a pathway involved in macrophage differentiation toward the M1 pro-inflammatory phenotype, which is based on reduced iron uptake and stabilization of IFN- γ R2 on the cell surface.

The experimental design is well thought out, the results are concisely described, and the figures appear convincing. However, some important details are missing thus hindering the positive assessment of the results.

The following issues need to be clarified:

1. Immunofluorescence data:

Lines 231-232, the author write: Immunofluorescence analysis revealed markedly higher levels of membrane IFN- γ R2 accumulation. – Fig. 6A.

Which membrane are they referring to? Is this the plasma membrane, as the text suggests? Enhanced staining of the cell perimeter suggests the plasmamembrane. The authors are cautious in their description, but the meaning of their argument is clear and points to the plasma membrane surface. I assume that cells were permeabilized to detect Rab5, although this is not indicated in Methods. If that is the case, we cannot be certain which membrane or surface is being visualized without further analysis.

Furthermore, in lines 284-285, the authors write about Fig. 8: Immunofluorescence revealed increased cell surface IFN- γ R2 and decreased cytoplasmic IFN- γ R2 when the iron supply decreased. The same set of proteins, as in Fig. 6, is shown here, and the “surface level of the receptor” is clearly stated.

Also, lines 272-273 say: Moreover, M1 macrophages presented more IFN- γ R2 on the cell surface (Fig. 7E).

In all these cases important details about the procedure of cell staining are missing and the statement of surface level of IFN- γ R2 seems premature.

Answer 1-1: We are very grateful to your kind, careful review and comments. Now we have revised the manuscript to describe the method more clearly.

To investigate the internalization of IFN- γ R2 from the plasma membrane to intracellular compartments, previously reported double immunofluorescence staining of cultured cells was performed [Reference 1]. Briefly, cells were briefly washed with phosphate-buffered saline (PBS) and fixed in 4 % paraformaldehyde, permeabilized with mild condition of 0.03% Triton X-100 for 20 min, which was optimized through preliminary experiments. This mild permeabilization allowed effective antibody penetration while preserving the overall structure and integrity of the plasma membrane. After permeabilization cells were blocked with PBS containing 2 % BSA for 1h at room temperature. Then cells were stained with specific antibodies,

including anti-IFN- γ R2, anti-Rab5a at 4 °C overnight, followed by species-matched fluorescent secondary antibodies.

To further investigate the increased cell surface IFN- γ R2 in THP1-TMPRSS11E cells, the plasma membrane or endosome were isolated separately, and then the level of IFN- γ R2 was analyzed by western. The results were added in the revised manuscript.

Other issues:

1. Lines 122-124. The authors write: The cell surface TFR1 level in the mutant TMPRSS11E-transfected group was significantly greater than that in the wild-type TMPRSS11E-transfected group, reflecting the impaired cleavage of mutated TMPRSS11E.

In lines 134-135, they write: The surface TFR1 level in the inhibitor group was similar to that in the empty control plasmid pCMV-GFP-transfected group. It is not clear which data these conclusions are based on.

Answer 1: Thank you for your valuable comment. To improve clarity, we now revised the description in the Results section. Please refer to the revised manuscript.

The statements in Lines 122–124 are based on the western results presented in Fig. 2C, the cell surface full length TFR1 in lane 2 and lane 3 are compared. The statements in Lines 134–135 are based on the western results presented in Fig.2D, the cell surface full length TFR1 in lane 2 and lane 4 are compared.

2. The TMPRSS11E construct with the GFP tag, shown in Figure 2C, is not sufficiently described. The Methods section should include basic information about it, such as the role of the SEA motif, etc.

Answer 2: Thank you for your valuable comment. Now we have added a more detailed description about GFP-tagged TMPRSS11E construct in Figure 2C. The Methods section provide the basic information about the TMPRSS11E structure. The SEA domain is a conserved structural motif found in various membrane-bound serine proteases. All members of the HAT/DESC subfamily such as TMPRSS11D, TMPRSS11E, TMPRSS11A, TMPRSS11B and TMPRSS11F, possess the structurally identical stem region that is composed of a single SEA domain. The extracellular part of TMPRSS11E consists of a 120-amino acid SEA domain followed by the C-terminal trypsin-like serine proteinase domain, as shown in Figure 2C. It was supposed that the SEA domain more likely functions by orienting the active site cleft of TMPRSS11E toward plasma and/or extracellular spaces and away from the cell surface and/or the extracellular matrix. The SEA domain may also contribute to the adhesion properties of TMPRSS11E-expressing cells and might localize ‘shed’ TMPRSS11E in appropriate microenvironments. However, the detailed role of SEA domain until now is not clear [References 2-3].

3. It is worth to clearly state in the text that the “TMPRSS11E recombinant protein” is in fact its catalytically active fragment, and a reference should be made to the previous study of the authors in Carcinogenesis, where the construct is described in detail.

Answer 3: Thank you for your valuable suggestion. According to your suggestion, we have revised the text to clearly state that “TMPRSS11E recombinant protein” is in fact the catalytically active fragment. In addition, the revised manuscript now cited our previous publication which described the TMPRSS11E recombinant protein in detail [Reference 4].

4. The legend to Fig. 4A, B, C, D: what does “IHC staining” refer to? Fig. 4B1, there is a similar simple statement about “Immunofluorescence staining”. Please, indicate which proteins are visualized in these images.

Answer 4: Thank you for your good comment. According to your advice, IHC was replaced by immunocytochemistry. In addition, we have updated the figure legends to clearly indicate the specific proteins visualized in these images.

5. Lines 178-179, the author write: Histological analysis revealed neutrophil infiltration in the lung tissues from mice in the LPS. What data are they referring to?

Answer 5: Thank you for your valuable comment. According to your suggestion, now we revised the manuscript. LPS-induced pulmonary inflammatory mice model was performed. H&E staining of the lung tissues showed thickened alveolar walls, less alveolar cavity and inflammatory cell infiltration in the LPS group. It was also found that MPO activity was significantly elevated in the LPS-treated group compared to controls which reflect neutrophil accumulation in the lungs. We have now included the MPO assay data as Fig. S2C in the revised Supplementary material.

6. The description of the approaches used to modulate iron uptake is not sufficient. DFO is not identified as an iron chelator.

Answer 6: Thank you for your valuable comment. According to your suggestion, approaches used to modulate iron uptake was provided more in the revised manuscript. Ferric ammonium citrate (FAC) is used extensively in the food industry as an additive and in medicine to treat iron-deficiency anemia in humans. In addition, FAC has become a standard source of iron in numerous biological studies. Increased cytosol labile iron Fe^{2+} content was evident in cells treated with FAC. Opposite effects were observed for treatments with iron-chelating agent DFO [Reference 5]. Deferoxamine (DFO) is an approved by FDA to treat iron overload. Deferoxamine binds to iron and removes it from the bloodstream. It functions by binding free ferric iron (Fe^{3+}) with high affinity, forming a stable complex that prevents iron from participating in cellular processes [Reference 6]. DFO has a significant association with a reduction in serum ferritin levels [Reference 7-8].

7. The legend of Fig. 8, lines 696-698, suggests that cells preincubated with ammonium ferric citrate were stimulated with LPS in serum-free medium, while cells treated with DFO - in medium containing serum. This is an important difference, since serum contains LPS-binding protein (LBP) which facilitates stimulation of cell with this endotoxin.

Answer 7: Thank you for your careful comment. This is a very good question. We

appreciate the reviewer's review. To consider the potential influence of serum-derived factors including LBP and serum ferritin on the result of experiment, cells were cultured in either complete or serum-free medium as control groups. Serum-free media eliminate the variability associated with animal-derived sera. To analyze the effects of iron concentration on the IFN- γ R2 distribution, we prepared four experimental groups: cells were cultured in serum-free culture medium as the control group 1, in serum-containing complete culture medium as the control group 2. In parallel, to evaluate iron concentration on IFN- γ R2 distribution, complete medium supplemented with DFO to remove free ferric iron (Fe^{3+}) and decrease serum ferritin levels as group 3. Serum-free medium supplemented with FAC to increase the iron concentration as group 4. We have now revised the figure legend to clearly describe the presence or absence of serum in each condition.

8. What was the chemotype of LPS used in the studies? In Table S2 "lipopolysaccharides" are mentioned, but no further detail is provided.

Answer 8: Thank you for your careful comment. As you suggested, in revised Table S2 of the Supplementary Materials, LPS chemotype (*Escherichia coli* O55:B5) was provided. LPS from *Escherichia coli* O55:B5 is a form of lipopolysaccharide (LPS) extracted from wild-type S-form *E. coli* serotype O55:B5. It is commonly used to activate toll-like receptor 4 (TLR4) on leukocytes, eliciting inflammatory signaling in isolated cells and in vivo.

9. In Fig. 9, abbreviations such as DMT1 and STEAP appear but are not explained in the text.

Answer 9: Thank you for your careful review. According to your suggestion, we have revised the figure legend and provide description for these abbreviations in the revised manuscript. DMT1: divalent metal transporter 1; STEAP: six-transmembrane epithelial antigen of prostate. Transferrin binds Fe^{3+} to form transferrin/TFR1 complexes, which are endocytosed into cell. Subsequently, in the endosome, Fe^{3+} is reduced to Fe^{2+} by STEAP and transported by DMT1 to the cytoplasm.

10. Fig. s1B – lanes of the gel are not described, making the Figure difficult to interpret.

Answer 10: Thank you for your valuable comment. As you suggested, Fig. s1B – lanes of the gel are described now in the revised manuscript.

In this experiment, TMPRSS11E-Flag was overexpressed in transfected cells, and subsequently immunoprecipitation was performed by using anti-Flag antibody to isolate target protein TMPRSS11E-Flag and their binding partner proteins from cell lysates. For protein-protein interaction analysis, immunoprecipitation with normal IgG was performed as negative control, which is to help identify adventitious proteins so that false-positive interactions can be annotated.

Before moving to the mass spectrometry step, Western blot was performed to confirm that the Co-IP was successful. The result for Western blot was shown in Fig.S1A.

Next, co-immunoprecipitated samples were loaded and run mini gel. And gel lanes

were cut and digested to peptides prior to LC-MS/MS. Fig.S1B showed the Coomassie-stained gel.

References:

1. Abboud D, Abboud C, Inoue A, Twizere JC, Hanson J. Basal interaction of the orphan receptor GPR101 with arrestins leads to constitutive internalization. *Biochem Pharmacol.* 220:116013(2024).
2. Fraser BJ, Wilson RP, Ferková S, Ilyassov O, Lac J, Dong A, Li YY, Seitova A, Li Y, Hejazi Z, Kenney TMG, Penn LZ, Edwards A, Leduc R, Boudreault PL, Morin GB, Bénard F, Arrowsmith CH. Structural basis of TMPRSS11D specificity and autocleavage activation. *Nat Commun.* 16:4351 (2025).
3. Kyrieleis OJ, Huber R, Ong E, Oehler R, Hunter M, Madison EL, Jacob U. Crystal structure of the catalytic domain of DESC1, a new member of the type II transmembrane serine proteinase family. *FEBS J.* 274(8):2148-60(2007).
4. Li S, Chen Z, Zhang W, Wang T, Wang X, Wang C, Chao J, Liu L. Elevated expression of the membrane-anchored serine protease TMPRSS11E in NSCLC progression. *Carcinogenesis* 43: 1092-1102 (2022).
5. Halcrow PW, Kumar N, Afghah Z, Fischer JP, Khan N, Chen X, Meucci O, Geiger JD. Heterogeneity of ferrous iron-containing endolysosomes and effects of endolysosome iron on endolysosome numbers, sizes, and localization patterns. *J Neurochem.* 161(1):69-83 (2022).
6. Regis G, Bosticardo M, Conti L, De Angelis S, Boselli D, Tomaino B, Bernabei P, Giovarelli M, Novelli F. Iron regulates T-lymphocyte sensitivity to the IFN-gamma/STAT1 signaling pathway in vitro and in vivo. *Blood* 105: 3214-3221 (2005).
7. Cappellini MD, Scaramellini N, Leoni S, Motta I. Iron Chelation Therapy. *Adv Exp Med Biol.* 1480:361-370(2025).
8. Abril Carrillo MD, Natalia Michel Carrillo-López, Arantza Lizbeth García-Loera, Yenny Viviana Pinzón-Ramírez, Luis China MD. Comparative Efficacy and Safety of Iron Chelators in Sickle Cell Disease: A Meta-Analysis . *Blood.* 144, Supplement 1: 2505(2024).

Dear Referees,

We feel great thanks for your professional review work on our article. As you are concerned, there are several problems that need to be addressed. According to your nice suggestions, we have made extensive corrections to our previous manuscript. The questions raised by the referees are responded as follows, and the amendments were highlighted in red in the revised manuscript.

Reviewers' comments:

Reviewer #1 (Remarks to the Author):

The authors have addressed most of the previous comments and have improved the overall quality of the manuscript. I appreciate the inclusion of the number of repetitions and the enhanced discussion.

To facilitate the review process, I recommend that, in future revisions, the authors indicate the location of the newly added or revised text (e.g., specific line numbers or sections in the manuscript) where reviewer comments have been addressed. This would be more effective than copying the revised text solely into the response letter.

One main concern that remains is the densitometric evaluation. Although the authors stated in their responses that densitometric analysis was performed to confirm the robustness of the data and that these analyses were incorporated into the new figures, I was unable to locate them. Furthermore, the methodology section and figure legends do not include any explanation or mention of densitometry. Please clarify this point.

Additionally, please briefly incorporate into the manuscript text a rationale for the use of both mouse and rat models that you answered to Rev1 to avoid potential confusion. Finally, the manuscript should be carefully revised for grammar and typographical errors. For example:

- Line 328: “sTfR1” – please check context and formatting.
- Lines 376, 748, 795: incorrect or missing spaces.
- Line 715: missing word – possibly “analysis,” “blot,” or “blotting.”

Answer 1: We are very grateful to your kind, careful review and comments. We are very sorry for the careless in the last response letter. In this response letter we indicate the locations of all the newly added or revised text (specific line number in the manuscript) where reviewer comments have been addressed.

For densitometry analysis for western blot, now we add the analysis data in the supplementary dataset in Excel format. Graphs for quantification of western blots are added in revised Fig.1D, Fig.S2, Fig.3B-3C, Fig.8B2 and Fig.S4, respectively.

Original Fig.S2 and Fig.S3 were renamed as Fig.S3 and Fig.S5 in the resubmitted manuscript.

Furthermore, the methodology section and figure legends also added the description of densitometry analysis. Please check the methodology section line 486 in the revised manuscript.

Additionally, according to your suggestion, to avoid potential confusion, the rationale for the use of both mouse and rat models was incorporated in the manuscript text. Please check the line 166 and line 199 in the revised manuscript.

Finally, we are sorry for our careless mistakes. We have tried our best to polish the language in the revised manuscript. Thank you for your reminder.

Line 328: “sTfR1” was corrected to sTFR1 (now in line 351).

As suggested by the reviewer, the incorrect spaces in Lines 376, 748, 795 were corrected (now in line 399, 816, 863).

In line 715, missing word “blot” was added (now in line 776,781,786).

Reviewer #2 (Remarks to the Author):

I maintain my positive opinion regarding the topic, scope of the study, and the interesting results. The Authors successfully addressed some of my concerns and provided source immunoblots. I have some reservations about the way the data are presented, which in turn affects the overall clarity of the text.

For example, in response to my previous comment 1, the authors indicate in the letter which gel lanes should be compared in Fig. 2. Unfortunately, this information is still missing from the manuscript itself. Furthermore, neither the figure, its legend nor Methods specify how the individual proteins were detected in immunoblotting analysis. Among others, the information on the detection of GFP-TMPRSS11E with an antibody specific against the protease itself, not the tag (GFP), can facilitate the analysis of the immunoblots. The term 'western' is used to refer to immunoblotting analyses. Similarly, in the case of immunocytochemistry staining and immunofluorescent detection, it is not stated which antibodies were used for the analysis. To keep the Figure legends concise, these details can be included in the respective sections of Methods with reference do Supplemental list of antibodies. At present, all the data on antibodies used in immunoblotting are included in the file “Related WB”, and there is also a list of antibodies found in the Supplement; however, it does not seem sufficient, as it does not specify which antibodies were used for which type of analysis (WB, IF, ICH). Regarding my previous remark 6 – the explanation of the methodological approach is provided in the letter; however, the description in the manuscript remains brief, and as a result, unclear. Incorporating these revisions should improve the readability of the text.

Answer: We sincerely appreciate the valuable comments, and hope the correction will meet with approval. Once again, thank you very much for your suggestions.

Based on your comments, gel lanes should be compared in Figure. Now the graph representing the western blot quantification of the protein relative levels are added. Please check the revised Fig.S2, the quantification of western blots in Fig.2 was provided. The comparison information about Fig. 2C and Fig. 2D was also added in revised manuscript. Please check the line 122,128 in the revised manuscript.

Furthermore, in figure 1, 2 and 3, the primary antibody used to detect proteins in immunoblotting analysis were added, the information about the antibody used for detection of GFP-TMPRSS11E was also included. Please check the revised figures 1-3.

The term 'western' is corrected as western blot through the text.

And we also correct the manuscript to provide information about which antibodies were used for which type of analysis (WB, IF, ICH). Please check the supplementary table S1 and materials section for IHC and IF (line 498, line 507,515).

Regarding the previous remark 6 – the explanation of the methodological approach used to modulate iron uptake was provided in the previous letter; now according to your suggestion, the explanation of the methodological approach was incorporated in the revised manuscript to improve the readability of the text. Please check the line 314 in the revised manuscript.

Ferric ammonium citrate (FAC) is used extensively in the food industry as an additive and in medicine to treat iron-deficiency anemia in humans. In addition, FAC has become a standard source of iron in numerous biological studies. Increased cytosol labile iron Fe^{2+} content was evident in cells treated with FAC. Opposite effects were observed for treatments with iron-chelating agent DFO [Reference 1]. Deferoxamine (DFO) is an approved by FDA to treat iron overload. Deferoxamine binds to iron and removes it from the bloodstream. It functions by binding free ferric iron (Fe^{3+}) with high affinity, forming a stable complex that prevents iron from participating in cellular processes [Reference 2. DFO has a significant association with a reduction in serum ferritin levels [Reference 3-4].

References:

1. Halcrow PW, Kumar N, Afghah Z, Fischer JP, Khan N, Chen X, Meucci O, Geiger JD. Heterogeneity of ferrous iron-containing endolysosomes and effects of endolysosome iron on endolysosome numbers, sizes, and localization patterns. *J Neurochem.* 161(1):69-83 (2022).
2. Regis G, Bosticardo M, Conti L, De Angelis S, Boselli D, Tomaino B, Bernabei P, Giovarelli M, Novelli F. Iron regulates T-lymphocyte sensitivity to the IFN-gamma/STAT1 signaling pathway in vitro and in vivo. *Blood* 105: 3214-3221 (2005).
3. Cappellini MD, Scaramellini N, Leoni S, Motta I. Iron Chelation Therapy. *Adv Exp Med Biol.* 1480:361-370(2025).
4. Abril Carrillo MD, Natalia Michel Carrillo-López, Arantza Lizbeth García-Loera, Yenny Viviana Pinzón-Ramírez, Luis China MD. Comparative Efficacy and Safety of Iron Chelators in Sickle Cell Disease: A Meta-Analysis . *Blood.* 144, Supplement 1: 2505(2024).

Reviewer #1 (Remarks to the Author):

The authors have addressed the previous comments and improved the overall quality of the manuscript. In my view, the manuscript is now suitable for publication in Communications Biology.

Response: We sincerely appreciate your insightful comments and positive evaluation of our work.